# Circadian clock regulates hepatic polyploidy by modulating Mkp1-Erk1/2 signaling pathway

Hsu-Wen Chao[1,6], Masao Doi[1], Jean-Michel Fustin[1], Huatao Chen [1], Kimihiko Murase[1,2], Yuki Maeda[1], Hida Hayashi[1], Rina Tanaka[1], Maho Sugawa[1], Naoki Mizukuchi[1], Yoshiaki Yamaguchi[1], Jun-ichirou Yasunaga [3], Masao Matsuoka [3,7], Mashito Sakai[4], Michihiro Matsumoto[4], Shinshichi Hamada[5] & Hitoshi Okamura[1]

Liver metabolism undergoes robust circadian oscillations in gene expression and enzymatic activity essential for liver homeostasis, but whether the circadian clock controls homeostatic self-renewal of hepatocytes is unknown. Here we show that hepatocyte polyploidization is markedly accelerated around the central vein, the site of permanent cell self-renewal, in mice deficient in circadian *Period* genes. In these mice, a massive accumulation of hyperpolyploid mononuclear and binuclear hepatocytes occurs due to impaired mitogen-activated protein kinase phosphatase 1 (Mkp1)-mediated circadian modulation of the extracellular signal-regulated kinase (Erk1/2) activity. Time-lapse imaging of hepatocytes suggests that the reduced activity of Erk1/2 in the midbody during cytokinesis results in abscission failure, leading to polyploidization. Manipulation of Mkp1 phosphatase activity is sufficient to change the ploidy level of hepatocytes. These data provide clear evidence that the *Period* genes not only orchestrate dynamic changes in metabolic activity, but also regulate homeostatic self-renewal of hepatocytes through Mkp1-Erk1/2 signaling pathway.

[1] Department of Systems Biology, Graduate School of Pharmaceutical Sciences, Kyoto University, Kyoto 606-8501, Japan. [2] The Department of Respiratory Care and Sleep Control Medicine, Graduate School of Medicine, Kyoto University, Kyoto 606-8507, Japan. [3] Laboratory of Virus Control, Institute for Virus Research, Kyoto University, Kyoto 606-8507 Japan. [4] Department of Molecular Metabolic Regulation, Diabetes Research Center, Research Institute, National Center for Global Health and Medicine, Tokyo 162-8655, Japan. [5] Department of Pathology, Otsu City Hospital, Otsu 520-0804, Japan. [6]Present address: Department of Physiology, School of Medicine, College of Medicine, Taipei Medical University, Taipei 11031, Taiwan. [7]Present address: Department of Hematology, Rheumatology, and Infectious Diseases, Graduate School of Medical Sciences, Faculty of Life Sciences, Kumamoto University, Kumamoto 860-8556, Japan. Correspondence and requests for materials should be addressed to H.O. (email: okamurah@pharm.kyoto-u.ac.jp)

Time is an integral part of our life. Most organisms on Earth show daily cycles of physiology and behavior in harmony with light-dark cycles in the environment. Endogenous time is generated by a self-sustained molecular oscillator composed of a transcriptional–posttranscriptional feedback loop in which clock genes regulate their own transcription[1–3]. This core oscillator couples to cell metabolism and maintains proper rhythms in endocrine and metabolic pathways required for organismal homeostasis. Liver is particularly important as a central organ for glucose, lipids and nucleotides metabolism in the body[4–6]. The circadian clock rhythmically regulates genomic and epigenetic processes to anticipate and adapt to cycles of feeding/nutrition that have dramatic consequences on metabolic activity[7,8].

Since liver is the primary organ to which all nutrients and toxins are conveyed from the intestine via portal vessels, homeostatic renewal of hepatocytes is essential for its maintenance. Renewal occurs steadily and continuously, all hepatocytes being completely replaced within 2 years[9]. Genetic tracing studies provide evidence that homeostatic renewal operates via mature hepatocytes self-replication[10,11], fueled possibly by recently identified axin2+ stem cells located around the central vein of the hepatic lobule[12], and by Lgr5+ stem cells in neighboring region of the portal vessels[13,14]. However, the circadian mechanism underlying the homeostatic cell renewal is totally unknown, unlike the relatively well established mechanism of metabolism regulation by the clock[7].

Here, we investigated the role of the circadian clock in homeostatic renewal of hepatocytes by generating Period deficient (Per-null) mice, which lack all three Period gene homologs (Per1, Per2, and Per3). Period genes function as the central cogs of the circadian clock[1,2,15]. We discovered that the arrhythmic Per-null liver is characterized by massively accelerated hepatocytes polyploidization (average 16n to 32n) in midlobular to centrilobular regions surrounding the central vein, where the permanent renewal of cells occurs[12].

Cells usually contain two pairs of chromosomes (2n, diploid) but can sometimes have higher ploidy levels[16,17]. In general, eukaryotic organisms prefer a diploid complement of chromosome as it enables sexual reproduction and genetic recombination, but polyploidy is surprisingly common in nature, especially in plants, fungi, insects, fishes and amphibians[18]. In comparison, higher vertebrates do not appear to tolerate polyploidy very well, such that germline polyploidy drives embryonic lethality and accounts for 10% of spontaneous abortions in humans[19,20]. Somatic polyploidy, however, commonly occurs in specific tissues, and hepatocytes have been extensively studied as a model of polyploidy over 100 years[21]. Hepatocyte polyploidy is age-dependent[22,23] and is commonly observed across mammalian species[24]. In recent years, several factors including E2F1/7/8[25], insulin[26] and miR-122[27], have been reported to regulate hepatic polyploidy during liver development. However, why and how hepatocytes develop polyploidy is still a mystery waiting for further work in addressing this remarkable biological phenomenon.

By analyzing the over-polyploid liver of Per-null mice in vivo and in vitro, we found that the absence of Periods causes constantly high levels of the mitogen-activated protein kinase phosphatase 1 (Mkp1). Increased Mkp1 activity in self-renewing hepatocytes in turn inactivates the extracellular signal-regulated kinase (Erk1/2) in the midbody during cytokinesis, resulting in abscission failure, ultimately leading to increased polyploidization. These findings indicate that the circadian clock genes not only orchestrate daily metabolic changes in hepatocytes, but also regulate homeostatic hepatocyte self-renewal through Mkp1-pErk1/2 signaling pathway.

## Results

**Over-polyploid hepatocytes in Per-null mice.** We generated Per-null mice by crossing Per1-knockout[28], Per2Brdm1-mutant[29], and Per3 knockout[30] mice (Supplementary Fig. 1a). As expected, Per-null mice completely lack circadian locomotor, eating and drinking rhythms in constant darkness, whereas day-night rhythms were observed under light-dark (LD) cycles (Supplementary Fig. 1b) due to the masking effect of the environmental lighting cycle. Per-null mice did not show any prominent abnormality in development when fed a normal diet under standard LD cycles (Supplementary Fig. 1c,d).

We performed a systemic histological survey of tissue morphology by comparing a series of conventional hematoxylin/eosin (HE)-stained sections from brain, lung, heart, kidney, liver, intestine, colon, pancreas, spleen, adrenal, testis, and skin samples from Per-null mice with those from wild-type (WT) mice (Supplementary Fig. 2). Among all tissues examined, a marked difference between genotypes was found only in the morphology of the liver with the appearance of macronucleated hepatocytes in Per-null liver (Fig. 1a,b).

This increase in nuclear size was zone-specific in Per-null liver: enlarged cells with a giant nucleus accumulated round the central vein (CV) but were rare around portal vessels (PV) in hepatic lobules (Fig. 1c). The analyses of the zonal distribution of polyploid cells in hepatic lobules using alternative histological criteria[31,32] found that the acceleration of polyploidization occurred in hepatic layers closer to the CV (3–9 out of 15 subdivisions) excluding its nearest layers (1–2) (Fig. 1d). According to the classical zonation of hepatocyte lobules, these cells locate in the centrilobular or midlobular zones, and thus, we describe these cells as centro-midlobular hepatocytes (CMH), distinct from PH (periportal hepatocytes).

Further analysis verified that Per-null liver had larger CMH nuclei ($>100\,\mu m^2$) compared to WT, for both mononucleated and binucleated cells (Fig. 1e), but PH were not significantly different (Fig. 1f). Developmentally, the genotype difference of enlarged nuclear size in CMH became gradually evident around 4–6 weeks of age in Per-null mice (Fig. 1g). By using flow cytometry, we found that the increased nuclear size was accompanied by increased ploidy levels: already at 3 weeks of age, Per-null liver was mostly populated with tetraploid (4n) hepatocytes (Fig. 1h, i), while diploid (2n) hepatocytes were predominant in WT. At 12 weeks of age, polyploidy of Per-null hepatocytes progressed further, with a main 8n population among 16n and 32n hepatocytes, while WT hepatocytes remained at 4n (Fig. 1i). These observations clearly demonstrate that polyploidy is markedly accelerated in hepatocytes in Per-null mice.

Since an increase in liver polyploidy can originate from the accumulation of mononuclear hepatocytes with polyploid nucleus (i.e. nuclear ploidy) and/or that of binuclear cells (cellular ploidy)[33,34], we sought to refine our analysis. Cytometrical analysis of liver sections by β-catenin/Hoechst-33342 immunohistochemistry, quantifying nuclear size and number per cell, revealed that both nuclear and cellular polyploidization are accelerated in Per-null CMH, characterized by a highly significant increase in 8n binuclear ($2 \times 8n$) cells as well as 16n mononuclear hepatocytes in the centrilobular zone (Fig. 2a–e). The developmental onset of binucleation and polyploidization coincides, observable by 6 weeks of age to become significant by 8 weeks (Fig. 2f), suggesting their etiology is developmentally connected.

To identify which Period gene mostly dictates polyploidy, we examined hepatocytes in single and double knockout mice, and found that CMH from Per1−/−, Per2Brdm1/Brdm1, Per3−/−, and Per1−/−;Per2Brdm1/Brdm1 mice all showed intermediate nuclear size between those of Per-null and WT mice (Supplementary Fig. 3a,b). It is noteworthy that PH did not show any difference in

all genotypes (Supplementary Fig. 3c). These findings indicate all three *Period* genes contribute to increased zone-specific polyploidization in hepatocytes.

**Abscission failure in *Per*-null hepatocytes**. Polyploid cells can arise from three general mechanisms, (i) cell fusion, (ii) cell development program known as endoreplication, and (iii) cell

defects that can induce an abortive cell cycle[16]. Polyploidy in hepatocytes is thought to result of the third mechanism[33,34]. We thus analyzed the entire course of cell division by time-lapse microscopy using cultures of primary hepatocytes from *Per*-null mice of 3 weeks of age, when polyploidy starts to develop (Fig. 3a, b and Supplementary Fig. 4a-d, Supplementary Movie 1). In both *Per*-null and control WT hepatocytes, mitosis successfully

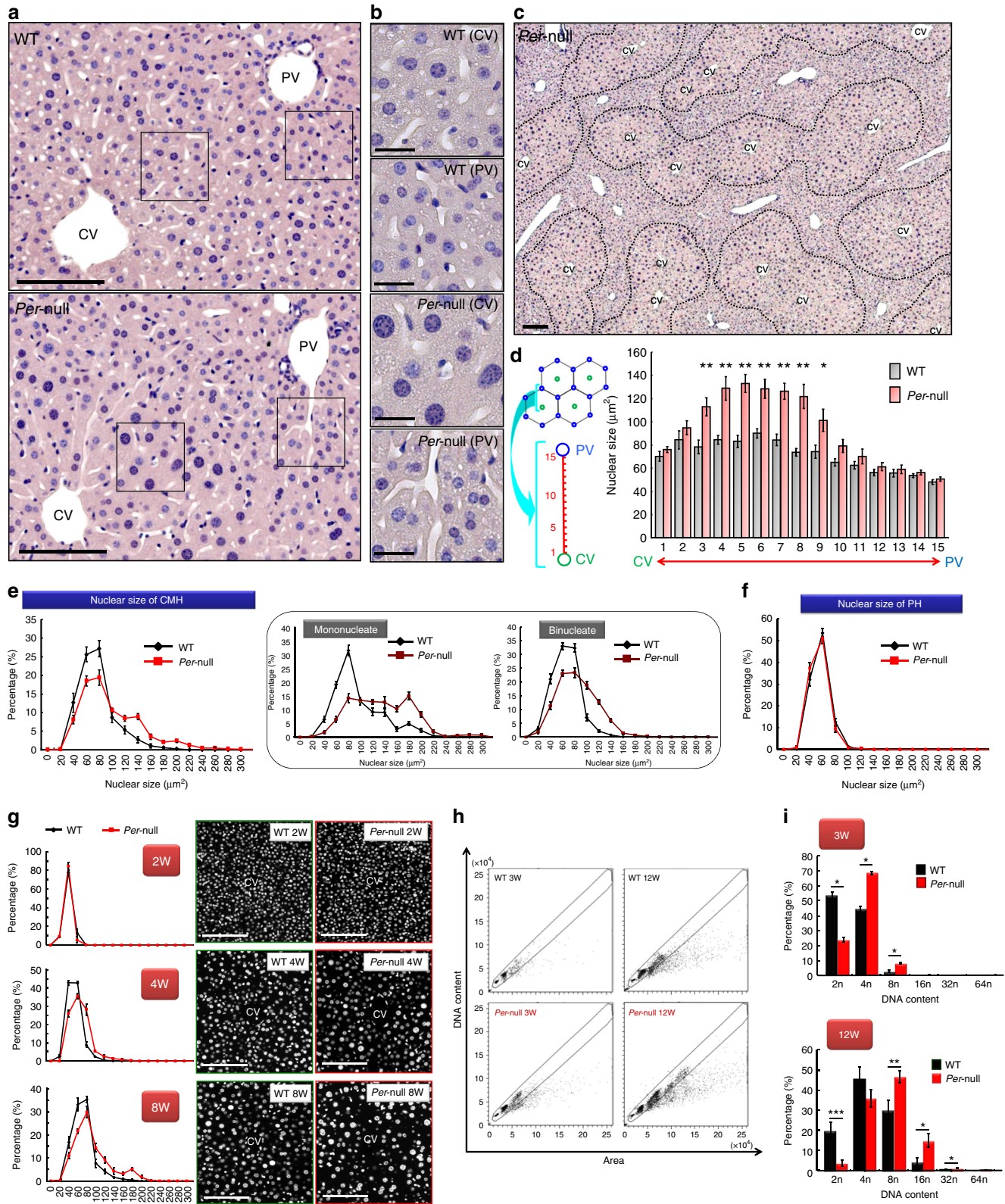

proceeded with no apparent abnormality, and no differences were observed in cytokinesis until the contractile ring formation (Fig. 3a and Supplementary Fig. 4e). In contrast, we observed a defect at the very last step of cytokinesis in *Per*-null hepatocytes: the separation between the two daughter cells (abscission) did not succeed, and instead the nascent daughter cells, still connected by their cleavage furrow, fused back together to form a binuclear cell (Fig. 3b). This abscission failure was rarely found in WT (5.30%: 28 cases among 526 hepatocytes), but dramatically increased in *Per*-null liver (17.5%: 126 cases among 717 hepatocytes) (Fig. 3c and Supplementary Fig. 4f).

While abscission failure can understandably produce multi-nucleated cells, it is unclear whether it can also produce cells with a single macronucleus. After abscission failure, cultured *Per*-null hepatocytes were continuously traced (Supplementary Fig. 4g), leading to the observation of a cell with a single 4n nucleus developing into an 8n after abscission failure (see Supplementary Movie 2). First, the cell with a 4n (based on Hoechst staining) nucleus entered into S-phase to become 8n, then progressed to M-phase but failed abscission, resulting in a binuclear hepatocyte (4n × 2). This cell then started a second S-phase (8n × 2), went into a second M-phase with both nuclei fusing, performed cytokinesis successfully, and then became two mononucleated hepatocytes each with a single enlarged nucleus (8n × 1; 2 cells). Thus, two cell cycles, the first failing abscission to produce binuclear cells, lead to polyploid hepatocytes with a single enlarged polyploid nucleus (Supplementary Fig. 4g, Supplementary Movie 2). A similar mechanism has been previously reported to occur in insulin-induced polyploidization[26,35]. Occurrence of abscission failure in *Per*-null hepatocytes regardless of the previous ploidy level (inferred by nuclear size) suggests that this abscission failure is the cause of the extraordinary polyploidization observed in vivo (2 × 8n, 1 × 16n, 1 × 32n).

**pErk1/2 is downregulated in *Per*-null mice.** We next proceeded to identify a potential molecular mechanism underlying the abscission failure and polyploidization in *Per*-null hepatocytes. We first checked the insulin signaling cascade in WT and *Per*-null cultured hepatocytes, since it has been implicated in weaning-evoked hepatocyte tetrapolidization[26] (Supplementary Fig. 5a). Interestingly, we found that phosphorylation of MAP kinase (Erk1/2) was significantly decreased in *Per*-null hepatocytes (Supplementary Fig. 5b,c). We then examined phosphorylated Erk1/2 (pErk1/2) across circadian time in vivo in WT and *Per*-null liver (Fig. 4a), and found consistently low levels of pErk1/2 in *Per*-null compared to WT at all timepoints. WT liver showed a

clear circadian rhythm, with a peak at CT0-CT4 and a trough at CT12-CT16, consistent with previous studies[36]. To localize the histological origin of this rhythm, we next performed pErk1/2 immunohistochemistry and found a robust oscillation specifically in WT CMH (Fig. 4b), except in pericentral stem cells (Supplementary Fig. 6a–e). pErk1/2 began to rise at CT20 in the cytosol, peaked at CT0/CT4 in the whole cell but intense in the nucleus, and declined at CT8 to CT12/CT16 (Fig. 4b, higher magnification). On the other hand, it is important to note that we never observed clear pErk1/2-positive staining in *Per*-null liver (Fig. 4b), indicating the loss of the circadian pErk1/2 activity in the CMH of *Per*-null mice. The sublobular distribution of pErk1/2-positive cells in WT (Fig. 4b and Supplementary Fig. 6c) overlaps with that of over-polyploid cells in *Per*-null liver (Fig. 1c), where cell proliferation also occurs even in adults (Supplementary Fig. 6e–h; also see the Supplementary discussions for Supplementary Fig. 6), although at much lower rate than in developing liver in both WT and *Per*-null mice (Supplementary Fig. 6i). These data suggest that low pErk1/2 activity in CMH is involved in polyploidy formation of *Per*-null mice.

To gain insight on the role of pErk1/2 in polyploidy development, we next examined the subcellular localization of pErk1/2 during abscission and found that pErk1/2 was concentrated in the midbody (Fig. 4c). Notably, in *Per*-null hepatocytes, the intensity of pErk1/2 at the midbody was significantly decreased, and relatively longer intercellular bridges between daughter cells were observed (Fig. 4c, d). To test a causal relationship between depletion of pErk1/2 at the midbody and abscission failure in *Per*-null hepatocytes, we pharmacologically inhibited phosphorylation of Erk1/2 with the Mek inhibitor U0126[37] in cultured WT hepatocytes. We observed longer intercellular bridges extending from the midbody in U0126-treated WT hepatocytes, comparable to *Per*-null hepatocytes (Fig. 4c, d). Moreover, time-lapse image analysis demonstrated that U0126 could dose-dependently increase the ratio of abscission failure in WT hepatocytes (Fig. 4e). Altogether with a report on Hela cells showing that inactivation of Erk1/2 inhibits disconnection of daughter cells during abscission[38], our results support the conclusion that abscission failure of *Per*-null hepatocytes is caused at least in part by pErk1/2 depletion.

**Constant upregulation of Mkp1 in *Per*-null mice.** We next examine the mechanisms of the downregulation of pErk1/2 in *Per*-null liver. This is likely to arise from a decrease in kinase activity and/or an increase in phosphatase activity. Mek1/2 are known dominant kinases phosphorylating Erk1/2, but we

**Fig. 1** Zone-specific polyploidization in *Per*-null mouse liver. **a–c** A representative HE-stained image of liver in WT and *Per*-null mice (male, 12 weeks old; $n$ = 6 mice for each group). High magnification photos of indicated areas of **a** are shown in **b**. Wide view of HE-stained *Per*-null mouse liver (**c**) shows clear zone-specific difference of hepatocytes. Hepatocytes encircling the central vein (CV) have larger nuclei than those encircling the portal vessels (PV). Dotted line borders between the two groups. **d** Zone-specific nuclear sizes stained with Hoechst-33342 of WT and *Per*-null mice. The bar graph shows the distribution of variant hepatic nuclear size (nuclear square: $\mu m^2$) along with the CV-PV axis. The CV-PV axis was subdivided into 15 parts (left schema), and the less number indicates the region closed to central vein region. The traditional 3-zone classification of hepatic lobule divides CV-PV axis into three equal parts: namely, centrilobular zone corresponds to 1–5, midlobular zone to 6–10, and periportal zone to 11–15. Note significant increase of nuclear diameter in *Per*-null hepatocytes than those of WT was observed from part 3 to 9, which distributes both centrilobular and midlobular zones, and thus, we describe these cells to centro-midlobular hepatocytes (CMH) separating from periportal hepatocytes (PH). The results were generated from five mouse livers for each group, and three different lobules were analyzed in each liver. **e** Frequency distributions of nuclear sizes of WT and *Per*-null CMH (**e**) and PH **f** ($n$ > 2500 cells from 5 mice for each group). **g** Developmental changes of nuclear sizes of CMH in WT and *Per*-null liver. The size of Hoechst-33342 stained nuclei was plotted against frequency distributions ($n$ > 2000 cells from 3 mice for each group). Flow sorting of single hepatocytes on the basis of DNA content against cellular area shows dot plot graphs **h** for gating criteria and histograms **i** of Hoechst-33342 stained nuclei, showing the DNA content of WT and *Per*-null hepatocytes at 3 weeks ($n$ = 3) and 12 weeks ($n$ = 4) of age. Data are representative of at least three independent experiments. Two-tailed unpaired Student's $t$-test with Welch correction was applied to **d**; Two-way ANOVA with Bonferroni's post-test was used to (**i**). Values represent the mean ± SEM, *$P$ < 0.05, **$P$ < 0.01, ***$P$ < 0.001. Scale bars, 100 μm in **a**, **c**, **g**, 20 μm in **b**

observed neither daily rhythms nor reduction of their amounts in *Per*-null liver, at mRNA (Supplementary Fig. 7a), protein, or phosphorylation levels (Supplementary Fig. 7b). On the other hand, by screening Map kinase phosphatase (Mkp) family genes expressed in the liver by quantitative RT-PCR (Supplementary Fig. 7c), we found that only *Mkp1* showed daily changes in WT

and was constantly upregulated in *Per*-null mice. Since two independent earlier reports[36,39] also showed diurnal rhythm of *Mkp1*, we decided to perform further characterization of the circadian expression of *Mkp1* in the liver of both genotypes.

In WT, hepatic *Mkp1* mRNA showed a robust rhythm with a peak at CT12 and a trough at CT0, but constantly high at all

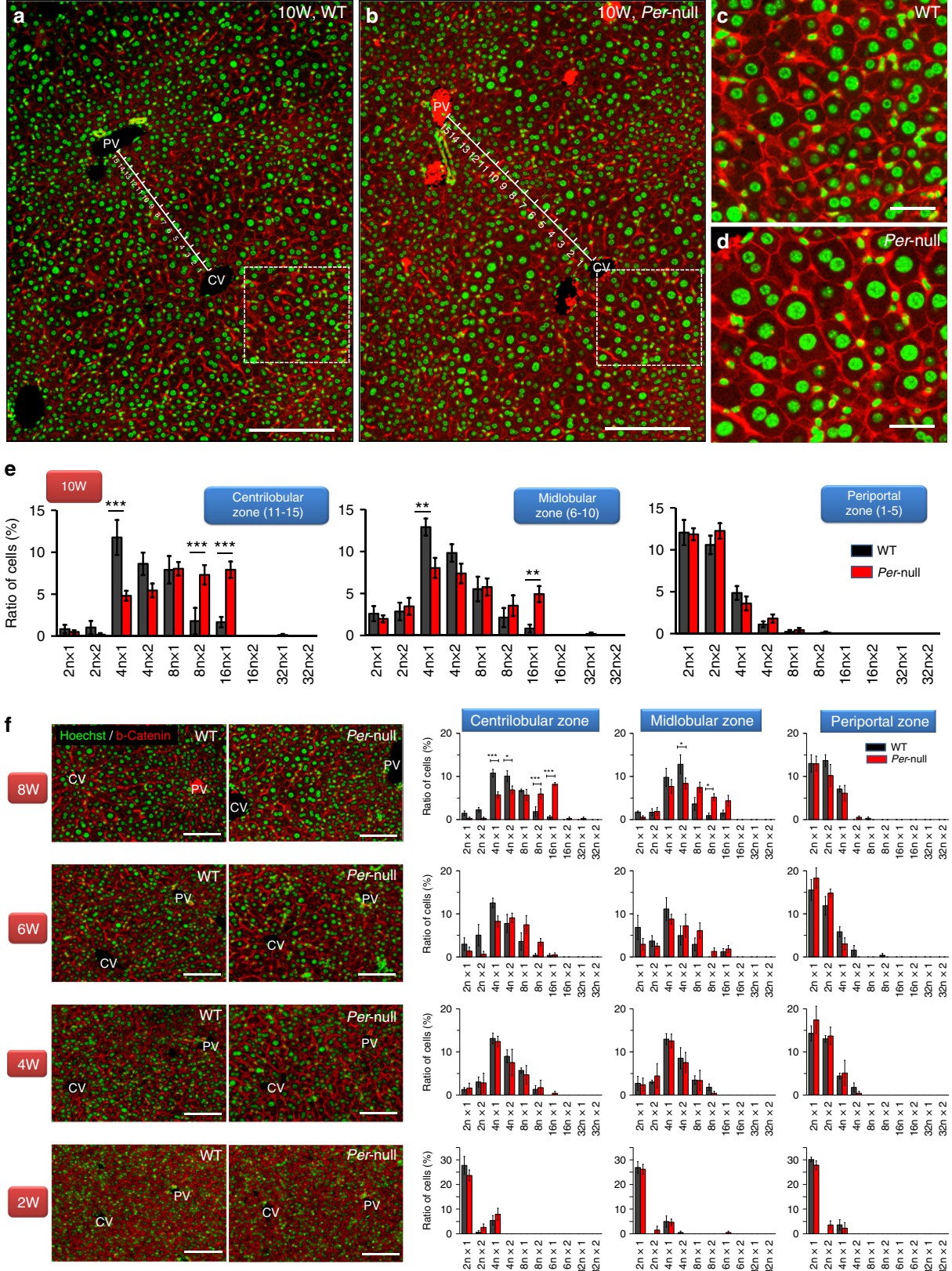

times in *Per*-null (Fig. 5a). In line with this transcript rhythm, Mkp1 protein levels fluctuate in a circadian fashion with a peak at CT12 and a trough at CT0 in WT, and constantly high in *Per*-null mice (Fig. 5b). Based on the observed negative correlations between Mkp1 and pErk1/2 abundance in WT and *Per*-null mice (Fig. 5b), published genome-wide chromatin immunoprecipitation sequencing (ChIP-seq) data for Clock and Bmal1[40,41] (Supplementary Fig. 8), and a ChIP qPCR assay that demonstrated the circadian time-specific recruitment of Per1 to the E-box enhancer element in the *Mkp1* promoter (Fig. 5c), we propose that the clock gene-controlled protein Mkp1 regulates phosphorylation of Erk1/2 in the liver.

**Mkp1 regulates the abscission failure and polyploidy**. To address whether the elevated Mkp1 expression is involved in abscission failure in *Per*-null mice, we treated *Per*-null hepatocyte cultures with BCI (CID 6419844/CHEMBL1241589)[42] (Supplementary Fig. 9a), an allosteric inhibitor of Mkp1, and found that the high incidence of abscission failures normally observed in these cells was dose-dependently decreased (Fig. 5d). Furthermore, to verify the specific effect of Mkp1 on abscission, we infected WT hepatocytes with lentiviral expression vectors driving Mkp1 overexpression (Supplementary Fig. 9b, c). We found that hepatocytes overexpressing Mkp1 tended to maintain intercellular bridges for a longer time during abscission and showed an increased abscission failure ratio (Fig. 5e, f and Supplementary 9d; see also Supplementary Movie 3). These findings demonstrate that the clock-controlled Mkp1 regulates the abscission of hepatocytes via pErk1/2 in the midbody.

Since these findings strongly suggest that polyploidy level is controlled by the Mkp1-pErk1/2 signal pathway, we directly challenged this hypothesis by pharmacological manipulations. When we applied U0126 to inhibit phosphorylation of Erk1/2, the nuclear size of WT hepatocytes was dramatically increased (Fig. 6). In contrast, when we applied BCI, the nuclear size of *Per*-null hepatocytes was decreased to a level comparable to WT (Fig. 6). These data demonstrate that nuclear ploidy can be controlled by manipulating the Mkp1-pErk1/2 pathway.

## Discussion

Cells with more than two paired sets of chromosomes are referred to as polyploid cells. These cells arise from normal diploid cells through yet unknown mechanisms. A paucity of animal models has so far hampered molecular characterization of cell polyploidization in mammals. In the course of a histological survey of the *Period* deficient mice, we found a massive accumulation of polyploid cells in their liver. Centrilobular hepatocytes exhibit particularly high ploidy levels, 8n, 16n, and 32n. Our molecular characterization revealed that the absence of *Periods* impairs the Mkp1-mediated circadian control of the phospho-Erk1/2 activity. Impaired Erk1/2 activity in the midbody during cytokinesis leads

to abscission failure of self-renewing mature hepatocytes, and this eventually enhances polyploidy. Our work therefore unmasked a role for a clock-controlled Mkp1-mediated Erk1/2 pathway in hepatocyte cytokinesis and polyploidization.

Mkp1 is a founding member of the mitogen-activated protein kinase phosphatase (MKP) family, expressed in many tissues and involved in a variety of functions by regulating the activity of mitogen-activated protein kinases (MAPK)[43]. The MAPK pathway conveys signals from cell surface receptors to the genome, and regulates fundamental cellular processes such as proliferation, differentiation, motility, stress response, apoptosis, and survival[43]. Mkp1 is an immediate early gene expressed in many tissues, with roles in both innate and adaptive immunity[44] and in lipid metabolism[45] and obesity in mice[46] and humans[47]. Here our data provide evidence that circadian oscillatory gene *Periods* are active in proliferating hepatocytes and involved in governing the abscission step of cytokinesis by regulating Mkp1 which inactivates pErk1/2 in the midbody. These findings open a new dimension of clock gene function in cell biology: intracellular clock components directly regulate cytokinesis, and *Periods* ablation results in abscission failure through circadian dysregulation of Mkp1.

The midbody is indispensable for completing abscission[48]. The midbody (or Flemming body) is a transient structure, containing bundles of microtubules derived from the mitotic spindle, found in mammalian cells and present near the end of cytokinesis, just prior to the complete separation of the dividing cells. Aside from microtubules, the midbody also contains various proteins involved in cytokinesis, asymmetric cell division, and chromosome segregation. Erk activation plays an important role for the disconnection of intercellular bridges between daughter cells at midbody, since the pErk1/2 inhibition by U0126 activates Aurora-B-dependent abscission checkpoint, which inhibits the abscission processes[49] through activation of endosomal sorting complexes required for transport (ESCRT)[50]. We observed phosphorylation of Aurora B at Thr232 with decrease of pErk1/2 in liver in vivo of *Per*-null mice (unpublished observation), which suggests that the persistent activation of Aurora B in *Per*-null mice stabilizes intercellular canals, and thereby prevents the completion of abscission.

The identification of the Mkp1-pErk1/2 pathway involvement in cytokinesis enabled us to manipulate polyploid states pharmacologically: pErk inhibition by U0126 increased the nuclear size of WT hepatocytes, and Mkp inhibition by BCI reduced the nuclear size of *Per*-null hepatocytes to a level comparable to that of WT nuclei (Fig. 6). This may be invaluable as a tool to investigate the molecular mechanism of polyploidization induced by oxidative stress and chemical-induced DNA damage[51], and during cell cycle disruption[25,52], and to further the biological significance of polyploidy that accompanies aging[23], inflammation[53] and tumorigenesis[54,55].

**Fig. 2** Nuclear ploidy and binucleation: zone-specific distribution and its developmental change detected by double staining immunohistochemistry with cell membrane marker β-Catenin and nuclear marker Hoechst-33342. By applying the immunohistochemistry of β-Catenin, we analyzed the nuclear ploidy and multinucleation in WT (**a, c**) and *Per*-null (**b, d**) mice (10 weeks). (**c**) and (**d**) are high-power field photomicrographs of indicated areas of (**a**) and (**b**), respectively. We divide the liver lobule in 15 zones, as indicated by the lines, and counted the mononuclear 2n cells (2n × 1), binuclear 2n cells (2n × 2), mononuclear 4n cells (4n × 1), binuclear 4n cells (4n × 2), mononuclear 8n cells (8n × 1), binuclear 8n cells (8n × 2), mononuclear 16n cells (16n × 1), binuclear 16n cells (16n × 2), mononuclear 32n cells (32n × 1) and binuclear 32n cells (32n × 2) according to the nuclear size of 2n cells in each section. **e** We show the incidence (%) of each ploidy state, mononucleated and binucleated, in each zone (centrilobular, midlobular and periportal zone) for each hepatic lobe (*n* = 600 hepatocytes from 5 mice of each genotype), assuming the contribution of each zone to be equal. For developmental analysis **f**, we analyzed liver from 2, 4, 6, and 8 weeks old WT and *Per*-null mice (*n* = 150–360 hepatocytes form 3–5 mice of each genotype at each developmental stage), and analyzed the incidence of each polyploidy state, of mononuclear and binuclear cells, in each developmental stage. Data are analyzed by Two-way ANOVA with Bonferroni's post-test. All values represent the mean ± SEM. *$P < 0.05$, **$P < 0.01$, ***$P < 0.001$. Scale bars, 100 μm in (**a, b, f**) and 20 μm in (**c, d**). Note hepatocytes increase their polyploidization progressively age-dependently, except for periportal cells, and the prominent polyploidization and binucleation progressively accelerates in pericentral and midlobular zones

Dysregulation of circadian rhythm is likely to accelerate malignant transformation of various cells as reported previously in malignant lymphoma[54] and breast cancer[35]. Of particular interest is the fact that mitosis of polyploid hepatocytes has been tied to aneuploidy[16,56], which arises through mitotic defects, chromosome missegregation, or chromosome catastrophe, and may give rise to hepatocellular carcinoma. However, we found no obvious aneuploid "population" either in WT or in Per-null hepatocytes in our flow cytometric analysis as shown in Fig. 1h. Furthermore, we found none of the Per-null mice harbors

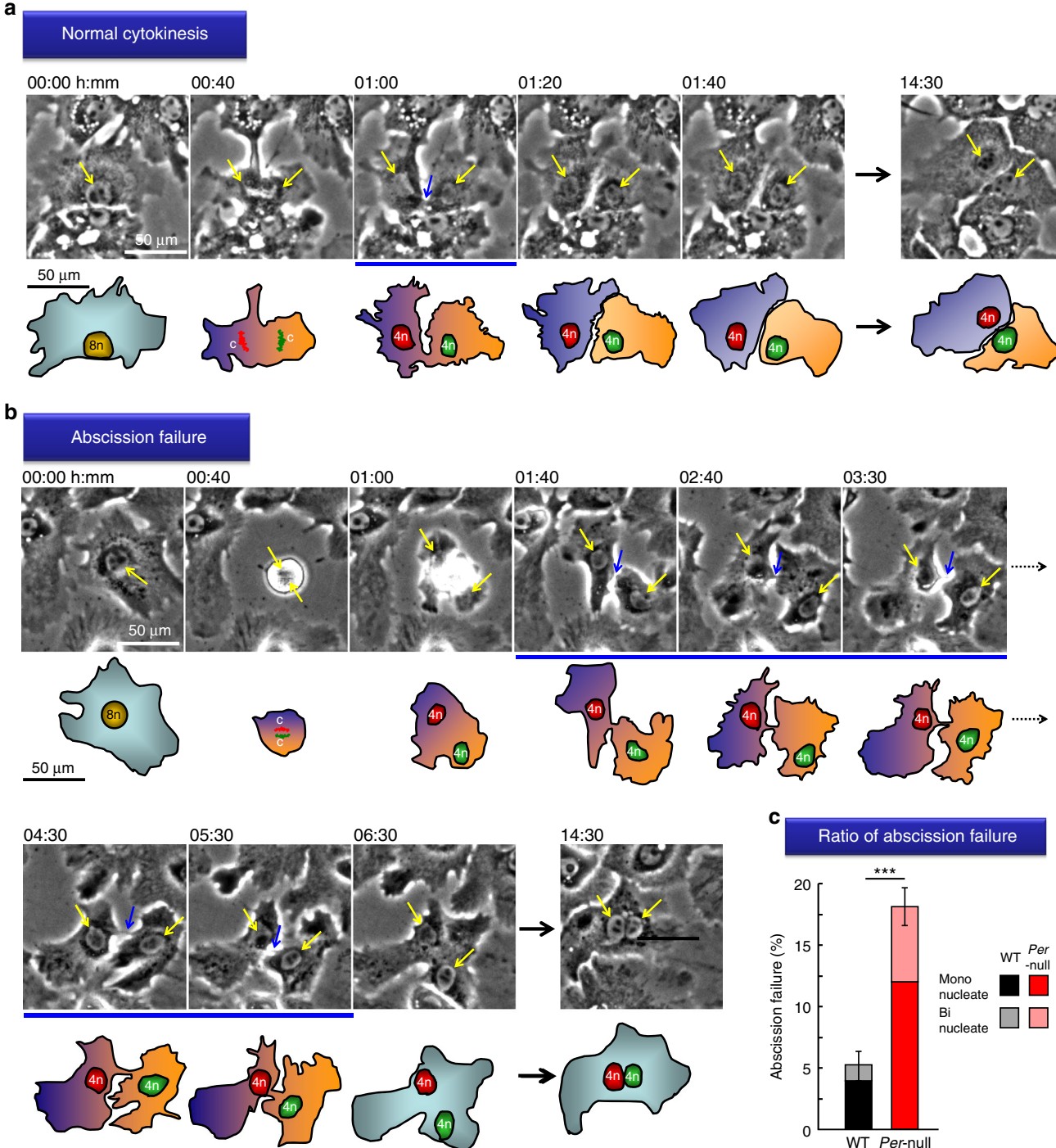

**Fig. 3** Genetic ablation of *Periods* triggers abscission failure. Representative time-lapse images of cultured hepatocytes showing **a** normal cytokinesis and **b** abscission failure. Yellow arrows indicate the localization of nuclei and chromosomes of dividing hepatocytes. Blue arrows exhibit intercellular bridges between daughter cells. Cartoons corresponding to time-lapse images are shown with the number of chromosomes (c) in the gametes (n). Appearance of intercellular bridge (blue underlines) between daughter cells was normally transient (~20−40 min) for normal cytokinesis but was extremely long (1−4 h) for abscission failure cells. **c** Quantification of the percentage of abscission failure. A marked increase of the abscission failure ratio in *Per*-null hepatocytes compared to that in WT. Data are representative of at least three independent experiments. Values represent the mean ± SEM, ***$P < 0.001$ by Student's unpaired *t*-test with Welch correction (WT, $n = 526$ cells; *Per*-null, $n = 717$ cells from five experiments for each group). All scale bars, 50 μm

hepatocellular carcinoma (HCC) until 6 months of age. Fu and colleagues recently demonstrated that chronic jet-lag increased incidence of hepatocellular carcinoma (HCC) in a recent paper[57] in both wild-type and clock gene-disrupted mice. However, according to their report, HCC was observed at aged mice: even the youngest mouse bearing HCC was 10 months old. Thus, our observations in *Per*-null mice probably indicate that additional

environmental events and/or aging, accompanied by genetic/ epigenetic alterations, are required for carcinogenesis in the liver.

Hepatic polyploidization occurs mainly during liver development. In rodents, hepatocytes are exclusively diploid (2n) in neonates, polyploidization starting after weaning. Up to 90% of rat/mouse hepatocytes become polyploid at adult age[24,58]. The current view in the field concerning the induction of

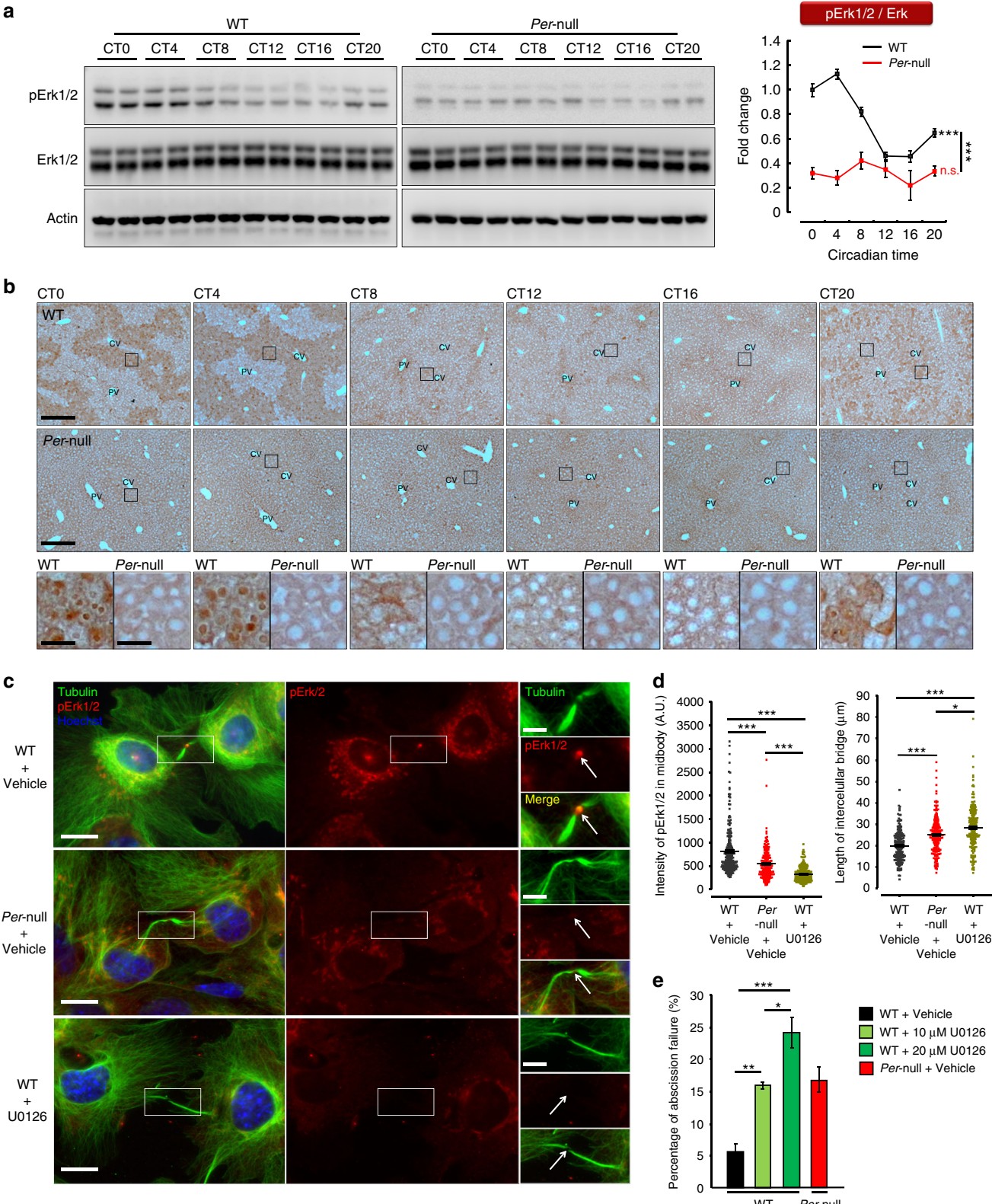

developmentally programmed liver cell polyploidization is that during weaning of mice, hepatocytes become first binucleated (cellular ploidy $2 \times 2n$) through a cytokinesis failure. Subsequently when a binucleated cell ($2 \times 2n$) enters another cell cycle and replicates its DNA and progress normally through mitosis and cytokinesis, two daughter cells are formed each containing a single enlarged nucleus ($4n$, nuclear ploidy). Thus, nuclear ploidy can be a consequence of cellular ploidy. This idea is in accordance with previous observations by Desdouets[59] that in wild-type mice after weaning, first binucleated hepatocytes appear in the liver, followed afterwards by mononucleated polyploid hepatocytes. Here we have demonstrated that this developmentally programmed polyploidization is accelerated in mice with deleted *Period* genes, resulting in the formation of more binucleated cells and mononucleated polyploid cells.

However, there is also some evidence that mononucleated polyploid hepatocytes may be derived directly from mononucleated diploid cells through a process called endoreplication[16,60]. Cells replicate their DNA without subsequent mitosis, so they progress through G1-S-G2 and back to G1 again. However, in our case of *Per*-null induced polyploidy, we cannot observe endoreplication during live cell imaging of hepatocyte culture: namely we cannot observe any instance showing the increase of nuclear size without progressing through mitosis. Endoreplication of hepatocytes has been described more under pathological conditions[60] such as non-alcoholic fatty liver disease or liver cancer.

The developmentally-regulated hepatocyte polyploidization is associated with high growth and high metabolic rate during the weaning period[24,26]. For example, hepatocyte polyploidization is increased in mice derived from small litters, where pups experience less competition and grow faster. Moreover, hepatocyte polyploidization is accelerated when the growth rate is stimulated by treating mice with triiodothyronine[25], growth hormone[24,61], or insulin[26] in contrast to the low rate of polyploidy in diet restriction[62], after thyroidectomy[61] and hypophyseactomy[24,61]. Interestingly, binuclear formation, which precedes mononuclear polyploidization, is delayed when weaning is delayed[63]. In addition to these developmental metabolic demands, the liver is exposed to circadian variation of absorbed foods from the intestine, which have profound impact on liver metabolism. It is known that Wee1 kinase, which phosphorylates and thereby inactivates the CDC2/Cyclin B1 complex to inhibit G2/M transition, is regulated by the clock[64,65], and is activated at dawn. These studies suggest that liver cells are actively differentiated in response to metabolic demand, and polyploidization is stimulated when energetic demands for proliferation compete with energetic requirements for differentiation in response to weaning or circadian cycles.

In conclusion, our study demonstrates that the Mkp1-pErk1/2 pathway is rhythmically activated in mature CMH hepatocytes predisposed to develop polyploidy and plays a critical role in the abscission into daughter cells. However, it should be kept in mind that cytokinesis is a complex phenomenon: in the abscission stage alone, dozens of regulators were identified[50], the *Per*-Mkp1-pErk1/2 pathway described here being only one among many causes of cytokinesis failure. Nonetheless, unmasking the molecular pathway underlying polyploidy sheds new light on the role of the clock in the day-to-day maintenance of liver mass.

## Methods

**Experimental animals**. All mice used for experiments were standard BALB/c strain with at least 10 times backcross breeding. Mice were maintained in 12-h light (~200 lux fluorescent light)/12-h dark cycle (LD) with food and water ad libitum. Before the analysis, mice were placed into an isolated light-dark box with LD cycles for at least 2 weeks to synchronize their circadian clock to the ambient light-dark cycle. Locomotor activity was recorded every 5 min with passive infrared sensor (Omron) and analyzed with Clocklab software (Actimetrics). To eliminate influence of external light on the internal clock of liver, in all experiments for circadian rhythmicity analysis, mice were exposed to constant darkness (DD) for 48 h before they were killed. Time is expressed as zeitgeber time (ZT), with ZT12 defined as the time of lights off in LD conditions, or circadian time (CT), with CT12 defined as the onset of locomotor activity in DD. To avoid the circadian influences to our data, 10–15 weeks old mice were used and killed at ZT8 for all experiments excluding special case of nuclear size measurement and hepatocytes primary culture. For nuclear size measurement at different developmental time, 2–20 weeks old mice were used for examination of nuclear size at indicated developmental stages, whereas, for hepatocytes primary culture, 21–25 days old mice were used. All animal experiments were approved by the animal experimentation committee of Kyoto University.

**Immunohistochemistry and measurement of nuclear size**. Specimens were fixed with 4% paraformaldehyde and embedded in paraffin-wax with standard protocol. Five-micrometer-thick sections were deparaffinized with xylene and ethanol and then antigen-retrieved by pressure cooking in Tris–EDTA buffer (pH 9.0) for 5 min, as described[66]. Sections were immersed in PBS containing 0.2% Triton X-100. For diaminobenzidine (DAB) labeling, sections were treated with 3% hydrogen peroxide for 30 min. Before blocking with 5% BSA and 5% FBS in PBS for 1 h, sections were treated with Super Block (ScyTek Lab) for 10 min and washed with PBS. Primary antibodies to β-Catenin (Cell Signaling, 9582), Ki67 (Abcam, ab16667), and phospho-Erk1/2 (Cell Signaling, 9101) were diluted in PBS containing 1% BSA, 1% FBS, and 0.1% Triton X-100. Following overnight incubations at 4 °C, sections were washed extensively with PBS containing 0.3% Tween 20 and incubated with secondary antibodies conjugated with biotin (1:500, Vector Labs) or Alexa fluorophores (1:500, Invitrogen) with the same condition as the primary antibody. For DAB labeling, sections were incubated with a pre-formed biotin-avidin-horseradish peroxidase complex (1:500, Vector Labs) at room temperature for 60 min, followed by wash with PBS in 0.3% Tween 20 and 50 mM Tris-HCl (pH 7.4) for 5 min twice sequentially at room temperature. Development of DAB was performed by incubation with 0.02% DAB (Wako) solution (0.02% DAB in 50 mM Tris-HCl, pH 7.4 with 0.001% $H_2O_2$) at room temperature for 5 to 10 min. Sections were mounted with entellan (Merck) after dehydration. For immunofluorescence labeling, sections were incubated with Hoechst-33342 (Molecule Probe) to mark the nuclei. Sections were mounted in Prolong Gold mounting medium with anti-fade reagent (Invitrogen) for image acquisition. For evaluation of sizes of nuclei and cells, Hoechst 33342 labeling and anti-β-catenin antibody staining were applied to outline nuclear and cellular morphology, respectively. Images were acquired by fluorescent microscope (Carl Zeiss, Axio Imager M2) equipped with 20X/ 0.3 EC Plan Neofluar objective and AxioCam MRm CCD camera on the resolution with 1388 × 1040 pixel. Images were subjected to ImageJ for nuclear and cell size quantification. For nuclear size measurement, the nuclei with round and oval morphology were analyzed. For the cell size measurement, only cells with round or oval nuclei were analyzed. Cell size was measured according to β-catenin signal. Data were obtained from three to five mice for each group. For

---

**Fig. 4** Zone-specific rhythmicity of pErk1/2 is suppressed in *Per*-null liver. **a** Immunoblotting of phospho-Erk1/2 (pErk1/2) and total-Erk1/2 (Erk1/2) in the liver at indicated circadian time (CT). pErk1/2 showed circadian rhythms in WT with a peak at CT4, but was constantly low in *Per*-null mice. Effect of time was significant in WT by One-way ANOVA. Two-way ANOVA reveals a significant difference between the two genotypes (WT, $n = 5$; *Per*-null, $n = 3$). **b** Immunohistochemistry of pErk1/2 at indicated times in liver. Prominent circadian change of pErk1/2 was found around CV region in WT but not in *Per*-null mice. High power fields show changes in nuclear pErk1/2 of CMH. **c** Immunofluorescence of pErk1/2 (red) during abscission in cultured hepatocytes counterstained with tubulin (green) and Hoechst-33342 (blue). Boxed areas were enlarged to show lowered pErk1/2 expression at the midbody of *Per*-null and U0126-treated WT cells. **d** Quantification of pErk1/2 intensity at the midbodies (left) and length of intercellular bridge between daughter cells from **c** (WT, $n = 204$ cells; *Per*-null, $n = 205$ cells; and U0126-treated WT, $n = 192$ cells; 3 experiments for each group). **e** U0126 dose-dependently increases abscission failure in cultured hepatocytes under time-lapse recording (WT vehicle, $n = 323$; WT 10 μM U0126, $n = 214$; WT 20 μM U0126, $n = 205$; *Per*-null, $n = 283$). Data are representative of at least three independent experiments. Two-way ANOVA (**a**) and One-way ANOVA (**d** and **e**) with Bonferroni's post-test were applied. All values represent the mean ± SEM. *$P < 0.05$, **$P < 0.01$, ***$P < 0.001$. Scale bar, 100 μm (**b**, low magnification), 20 μm (**b**, high magnification; **c**, low magnification), 5 μm (**c**, high magnification)

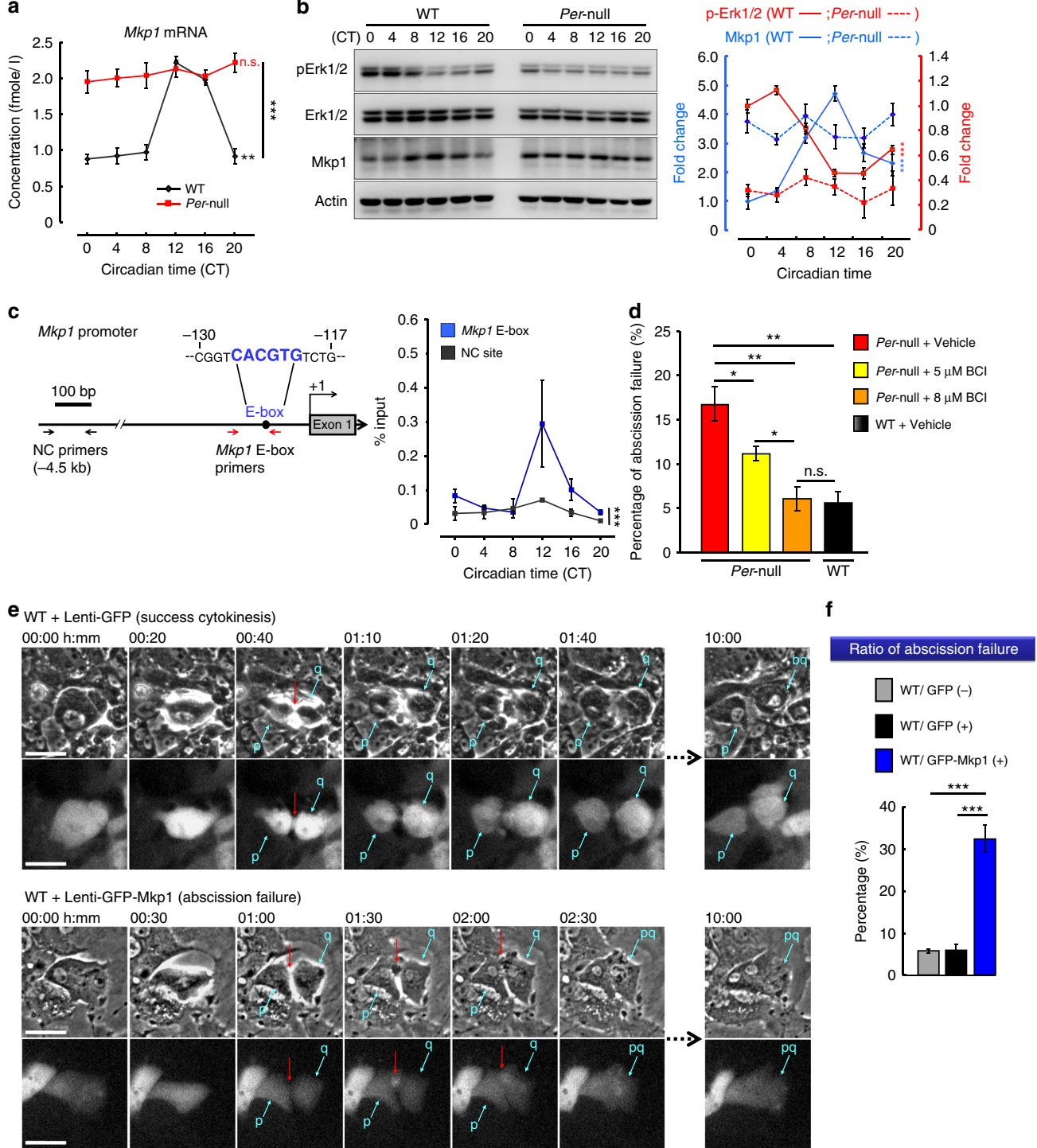

**Fig. 5** Upregulation of *Mkp1* in *Per*-null liver promotes abscission failure. **a** Rhythmic expression of *Mkp1* in WT liver but constitutively high in *Per*-null (WT, *n* = 5; *Per*-null, *n* = 3). **b** Immunoblotting and its quantitative analysis showing antiphasic expression of pErk1/2 and Mkp1 in WT and *Per*-null livers. (WT, *n* = 5; *Per*-null, *n* = 3). **c** Chromatin immunoprecipitation (ChIP)-assay of circadian Per1 binding on *Mkp1* promoter E-box. ChIP regions are schematically shown on *Mkp1* promoter. NC, negative control. **d** Mkp1 inhibitor (BCI) dose-dependently decreases the ratio of abscission failure in cultured *Per*-null hepatocytes (vehicle, *n* = 283; 5 μM BCI, *n* = 275; 8 μM BCI, *n* = 304) to normal levels (WT, *n* = 323). **e,f** Overexpression of Mkp1 increases the ratio of abscission failure in cultured hepatocytes. Representative images of cells infected with lentivirus carrying either Mkp1-GFP (Lenti-Mkp1-GFP) or GFP (Lenti-GFP) (**e** and Supplementary Movie 3), and quantification of abscission failure in indicated group (**f**). Red arrows indicate the intercellular bridge between two daughter cells (p and q) (WT/GFP-, *n* = 257, WT/GFP +, *n* = 209, WT/GFP-Mkp1 +, *n* = 189; 3 experiments for each group). Data are representative of at least three independent experiments. Two-way ANOVA (**a**–**c**) and One-way ANOVA (**d**, **f**) with Bonferroni's post-test were applied. All values represent the mean ± SEM. *P < 0.05, **P < 0.01, ***P < 0.001. Scale bar, 50 μm

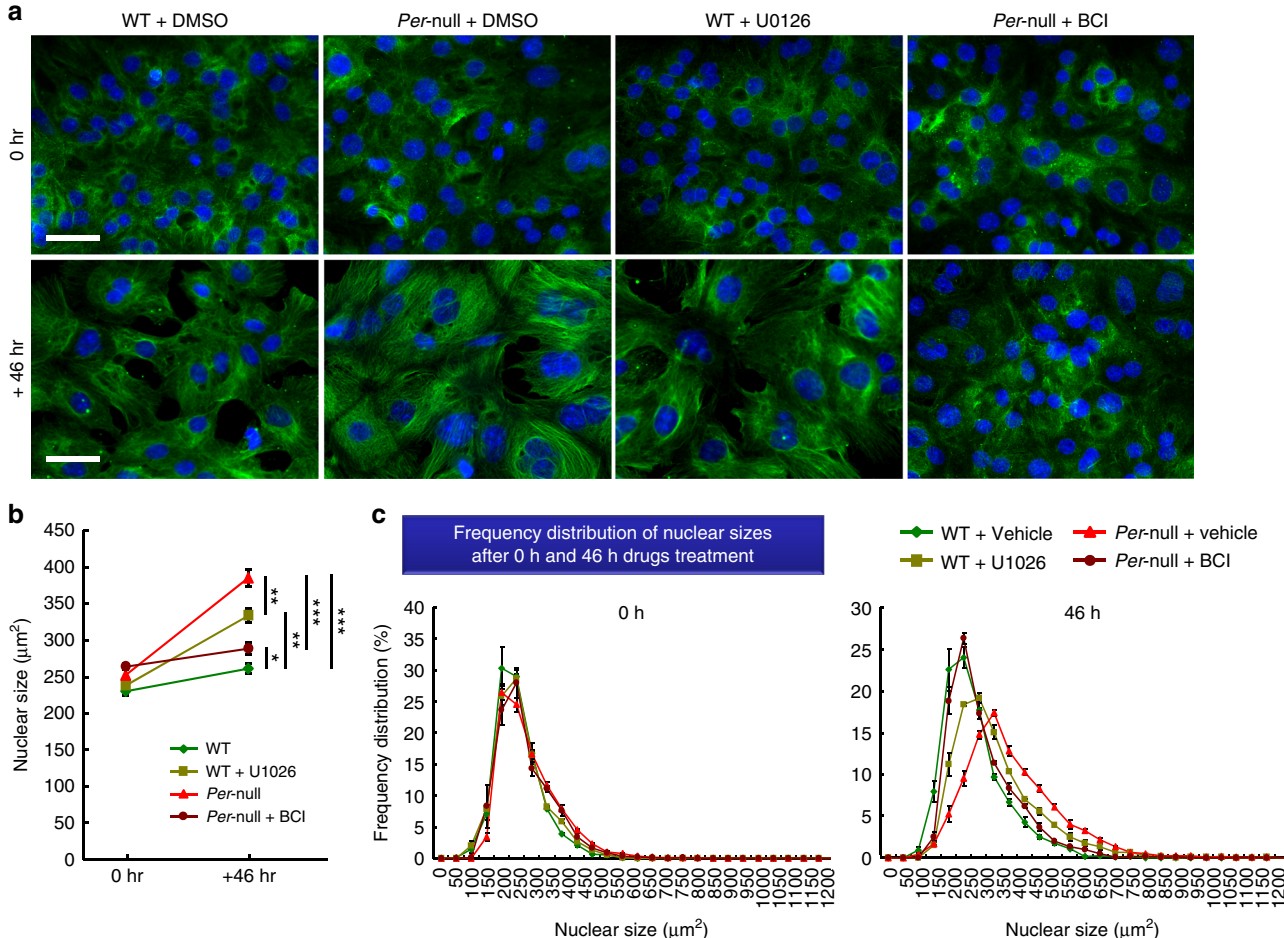

**Fig. 6** Pharmacological manipulations change the nuclear size of hepatocytes in culture. **a** Representative photos showing size change of nucleus after the treatment of Mek inhibitor (U0126) or Mkp1 inhibitor (BCI). Nuclei were stained blue with Hoechst-33342 (blue). Counterstained with tubulin (green). **b** Quantification of the alternation of nuclear size between 0 h or 46 h of treatment. Note that Mek inhibitor (U0126) treatment increases the nuclear sizes of WT hepatocytes to those in polyploidy *Per*-null mice. In contrast, Mkp1 inhibitor (BCI) reduced the nuclear sizes of *Per*-null hepatocytes to those in WT mice. **c** Frequency distribution of nuclear size of hepatocytes after 0 h and 46 h drug treatment. Data are representative of at least three independent experiments. Values are the mean ± SEM by Two-way ANOVA with Bonferroni's post-test., $n > 3000$ cells (three experiments). $*P < 0.05$, $**P < 0.01$, $***P < 0.001$. Scale bar, 100 μm

quantification, at least five different fields of the microscope were selected randomly from each mouse liver. Approximately 1500 to 3000 hepatocytes were analyzed for each animal. Mono- and binucleated hepatocytes were distinguished by comparing Hoechst-33342 and membrane-labeling images.

**Hepatocyte primary culture**. Primary mouse hepatocytes were isolated from 3-week-old mouse livers using a two-step collagenase I perfusion protocol[59,67,68]. Briefly, liver was perfused with $Ca^{2+}$ and $Mg^{2+}$ free Hank's balanced salt solution (HBSS) from inferior vein with a flow rate of 5 ml per minute for 5 min. An incision was made in the portal vein to let blood out of the liver. Liver was then perfused with 300 μg/ml collagenase I (Worthington Biochemical Corporation) in HBSS (Gibco) with a flow rate of 3 ml per minute for 15 min. After perfusion, liver was disrupted with scissors and dissociated by pipetting gently in 199 medium with 5% fetal bovine serum (FBS). Cell suspension was filtered through a cell strainer (100 μm²) and mixed with equal volume of 90% Percoll (Sigma) in HBSS. To collect live hepatocytes, the mixture was centrifuged at $100 \times g$ for 5 min, and the pellet was resuspended in 199 medium (Gibco) containing 5% FBS. Approximately $4 \times 10^5$ isolated hepatocytes were seeded on collagen-coated dishes (BD Falcon, 35 mm) and incubated at 37 °C with 5% $CO_2$. Medium was refreshed after 4 h of seeding. After 24 h of seeding, medium was replaced by 199 medium containing 10 mM HEPES (Gibco), 4.5 mg/ml glucose, 2 mM L-glutamine (Invitrogen), antibiotic-antimycotic (Gibco), 1 mM sodium-pyruvate (Gibco), 5 ng/ml sodium selenite (Sigma), 5 mM nicotinamide, 10 μg/ml transferrin (Sigma), 10 μM 3,39,5-triiodo-L-thyronine sodium salt (Sigma), 50 ng/ml recombinant mouse EGF (Invitrogen), 1μg/ml insulin (Sigma), 100 nM dexamethasone (Sigma), and 5% FBS. Medium was refreshed every 24 h throughout the culture time.

**Time-lapse recording**. The cytokinetic structures of hepatocytes were monitored in vitro by time-lapse microscopy (Carl Zeiss). Primary cultured hepatocytes isolated from the livers of 3-week-old mice were used since the proliferation ability of hepatocytes at this age is most vigorous and over 90% of hepatocytes are diploid and tetraploid (Fig. 1h,i). 24 h after seeding, images of hepatocytes were recorded live with a high resolution wide field inverted microscope Axio Observer Z1 system (Carl Zeiss). Cellular structures were visualized by phase contrast system with A-Plan 10 × / 0.25 dry objective lens and automatic exposure time. Hepatocytes were incubated in a chamber maintained at 37 °C and 5% $CO_2$ ($CO_2$/Temp Module S, Carl Zeiss). Images were taken at 10 min intervals for approximately 120 h by AxioCam MRm CCD camera. The optimal focal plane was set at beginning of each image session and adjusted by Definite Focus system with 10-s intervals throughout the image recording. The recorded images were analyzed via ImageJ software.

**Drug treatment of hepatocytes in culture**. Hepatocytes were treated with 20 μM U0126 (Promega) or 8 μM DUSP1/6 inhibitor/BCI (EMD Millipore) after 50 h of seeding for time-lapse recording. Culture medium with indicated inhibitor was refreshed every 24 h during cell recording. For analysis of the intensity of phospho-Erk1/2 in primary hepatocytes, 20 μM U0126 or 8 μM BCI was applied after 50 h of seeding, and cells were fixed after 10 to 15 h of drug treatment. For quantification of nuclear size of primary hepatocytes, 20 μM U0126 or BCI was applied after 50 h of seeding, and cells were fixed after 0 h or 46 h of drug treatment. For the group with 46 h drug treatment, culture medium with indicated drugs was refreshed every 24 h.

**Lentivirus infection in primary hepatocyte culture**. Mkp1 cDNA was amplified from liver cDNA by designed primer pairs (Mkp1 cDNA forward primer, 5′-GCAGAATTCATGGTGATGGAGGTGG-3′; Mkp1 cDNA reverse primer, 5′-GCTGCGGCCGCTCAGCAGCTTGGAGA-3′) and sub-cloned into pCSII-EF-MCS-IRES-hrGFP (a generous gift from Dr Hiroyuki Miyoshi, RIKEN, Tsukuba, Japan) with flanking EcoRI and NotI sites to generate Lenti-Mkp1 plasmid. Mkp-1-expressing or mock pCSII-EF-MCS-IRES-hrGFP plasmid was cotransfected with the packaging plasmids, pCMV-Δ8/9 and pVSV-G, into HEK293T cells using Polyethylenimine "Max" (Polysciences) to generate infectious lentivirus. Supernatant was collected 48 h after transduction. Viral particles obtained from two 10cm-dishes were concentrated by ultracentrifugation at $25,000 \times g$ for 2.5 h at 4 °C, and were infected immediately to $4 \times 10^5$ primary mouse hepatocytes in culture medium containing 4 g/ml polybrene. After overnight incubation, supernatant was replaced with fresh medium.

**Image acquisition and analysis**. In vivo and in vitro fluorescent images were analyzed quantitatively. We quantified the length of the β-tubulin-stained intercellular bridge between daughter cells. Intercellular bridge is a region with highly condensed microtubules, and it is defined as a transient structure with about 1 to 2 μm in diameter only appearing toward the end of cytokinesis and just prior to the complete separation of nascent daughter cells. Length of the brightly labeled tubulin bundle in the center of the intercellular bridge between nascent daughter cells was measured using ImageJ. The subcellular fluorescent intensity of pErk1/2 during mitosis was quantified by background-corrected line scans along the central spindle region or midbody using ImageJ. The integrated fluorescence intensity along this line was used as the estimated amount of specific protein in this region. Around 200 dividing hepatocytes were analyzed for each group from three independent experiments. All images were acquired with a fixed exposure time and condition in the same experiment. All quantification of fluorescence intensities was performed under raw 16-bit images, and data were analyzed by GraphPad Prism-5.0.

**Flow cytometry**. Hepatocytes were isolated and resuspended in ice-cold PBS at a density of $2 \times 10^6$ cells/ml. Cells were fixed with 70% ethanol at 4 °C overnight with gentle rotation. Cells were washed with PBS and stained with 1 μg/ml Hoechst-33342 at 4 °C for 60 min. DNA content of labeled cells was measured by flow cytometer (BD, FACSAria II). Acquired data were analyzed by BD FACSDiva software 6.0.

**Western blot**. Protein extracts were prepared from equal amounts of liver or cultured hepatocytes in Laemmli SDS buffer supplemented with fresh protease (Roche) and phosphatase inhibitors (Nacalai). Western blot was performed as described[69] with the following specific antibodies: anti-β–Tubulin (Sigma, T4026), anti-Mek1/2 (Cell Signaling, 9122), anti-phospho-Mek1/2 (Cell Signaling, 9121); anti-Erk1/2 (Cell Signaling, 9102); anti-phospho-Erk1/2 (Cell Signaling, 9101), and anti-Mkp1 (Santa Cruz, SC-1199). Hepatocyte protein extracts were prepared by homogenizing equal amounts of liver tissue in Laemmli SDS sample buffer. The homogenates were boiled at 95 °C for 10 min to denature protein, then centrifuged at 14,000 g for 10 min to remove cell debris. Proteins were resolved by 10% SDS-PAGE mini-gels using reagent from ATTO. Resolved proteins were transferred onto PVDF membrane (WSE-4050–4053 P plus membranes) by semi-dry protein transferring (ATTO; WSE 4040 HorizeBLOT 4M-R) at room temperature. Blots were blocked in TBS with 5% skim milk for 60 min and were followed by treatment with indicated antisera in TBS with 0.05% Tween 20. HRP-conjugated secondary antibodies were used for enhanced chemiluminescence (ECL) detecting system and blots were imaged by using Image Quant LAS 4000 (Fujifilm) with ECL-Prime (GE Healthcare Life Sciences). Full size images are presented in Supplementary Fig. 10.

**ChIP assay**. ChIP assay was performed as described[70] with modifications. Livers from mice were homogenized in PBS containing 2 mM disuccinimidyl glutarate (DSG; Pierce) and incubated at room temperature for 20 min. Formaldehyde was then added at 1% of final concentration and incubated for 5 min. Crosslink reaction was stopped by glycine (final concentration, 150 mM) on ice. The homogenates (~5 ml/liver) were mixed with 10 ml ice-cold 2.3 M sucrose buffer including 150 mM glycine, 10 mM HEPES pH 7.6, 15 mM KCl, 2 mM EDTA, 0.15 mM spermine, 0.5 mM spermidine, 0.5 mM DTT, and 0.5 mM PMSF and layered on top of a 5 ml cushion of 1.85 M sucrose buffer (with the same ingredients and including 10% glycerol). The mixtures were centrifuged at $105,000 \times g$ for 1 h at 4 °C in a Beckman SW28 rotor. The resultant nuclear pellets were resuspended in IP buffer (10 mM Tris-HCl pH 7.5, 150 mM NaCl, 1 mM EDTA, 1% Triton X-100, 0.1% sodium deoxycholate, 1 mM PMSF, protease inhibitor cocktail) and sonicated around 15 s for 80 times at 4 °C using a Bioruptor UCW-201TM apparatus (Tosho Denki). For each reaction, 10 μg fragmented chromatin (resuspended in 500 μl of IP buffer) was pre-cleared by incubation with 40 μl of protein A-agarose (Roche) for 2 h at 4 °C, which was followed by incubation with 2 μl anti-mPer1 rabbit antiserum (Millipore, #AB2201) at 4 °C overnight. 40 μl Protein A/G Plus-agarose (Santa Cruz) was then applied and the mixture was incubated for 1.5 h at 4 °C. Beads wash and DNA elution were performed as described[70]. Eluted DNA fragments were purified with Qiaquick Nucleotide Removal Kit (QIAGEN) and quantified by qRT-PCR using Mkp1 E-box primers (forward primer, 5′-TAGGCCGATGACGTCTTTG-3′; reverse primer, 5′-

CAAACAAACCGTTCTCCCCC-3′) and Mkp1 −4.5 kb/ negative binding site primers (forward primer, 5′-AGCCACCAAGTAGCAACAGC-3′; reverse primer, 5′-GATTCCTGGGTTGGACTGTG-3′).

**Statistical analysis**. Mice and cultured cells were randomly assigned for time-course study and drugs treatment respectively. Imaging fields were randomly selected during image acquisition. The sample size among experimental groups was kept as equally as possible. The experiments and analysis were conducted in a blind manner and replicated at least three times independently. For the in vivo studies, at least three mice were used for each group. Differences between variables were evaluated using the non-parametric Mann–Whitney test. GraphPad Prism 5.0 software was applied to produce the graphs and statistical analysis. We conducted experiments the intensity of pErk1/2 on the midbody (Fig. 4c), nuclear size along with CV-PV axis (Fig. 1d), change of the liver weight (Supplementary Fig. 1d), characterization of mitotic events in vitro and in vivo (Fig. 3c and Supplementary Fig. 4), and nuclear sizes (Supplementary Fig. 6a) to two-tailed unpaired Student's t-test with Welch correction. For the effect of time within a group (Figs. 4a, 5a–c and Supplementary Fig. 1a, 6d), the alternation of pErk1/2 intensity on the midbody and cellular dividing events after pharmacological treatment (Figs. 4d, e, 5d, f and Supplementary Fig. 9d), and the difference of mRNA expression level of Mkp family genes (Supplementary Fig. 7c) was statistically analyzed by One-way ANOVA. Two-way ANOVA was applied to DNA content analysis (Fig. 1i), the population of mono- and binuclear hepatocytes during liver development (Fig. 2e, f), the genotype effect on RNA, protein expression and liver weight during circadian time (Figs. 4a, 5a–c and Supplementary Fig. 1a, 6d) and developmental stage (Fig. 1c and Supplementary Fig. 6h, i), and the influence of pharmacological treatment at different timepoints (Fig. 6b). Bonferroni's multiple comparison test was applied for comparisons among multiple conditions following one-way or two-way ANOVA tests. Sample numbers and statistical results were indicated in the figure legends precisely. Data were presented as the mean ± s.e.m. or s.d., and P-values <0.05 were considered significant. P-values are represented as *P < 0.05, **P < 0.01 and ***P < 0.001.

**Data availability**. The data that support the findings of this study are available from the corresponding author upon request.

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

## Acknowledgements

We are grateful to Dr. Paolo Sassone-Corsi (UC Irvine) and Dr. David Weaver (U Massachusetts) for *Per1*-knockout mice and *Per3*-knockout mice, respectively. We also thank Okamura-lab members for assistances and discussions during the course of this study, particularly to Dr. Satoru Masubuchi for backcross breeding to standard BALB/c strain for *Per1*-knockout mice, Shoichi Urabe, Miho Yasuda and Shota Akazawa for *Per2*- and *Per3*-knockout mice. We also thank to Dr. Yi-Shuian Huang (Academia Sinica) for providing equipment in Taiwan, and to Dr. William J. Schwartz (University of Texas Dell Medical School) and Dr. Setsuya Fujita (Emeritus Professor of Kyoto Prefectural University of Medicine) for stimulating discussions. This research was supported by Core Research for Evolutional Science and Technology (JPMJCR14W3-CREST), Japan Science and Technology Agency (to H.O.), and Scientific grants from the Ministry of Education, Culture, Sports, Science and Technology of Japan (to H.O.), and grants from Takeda Science Foundation, Kobayashi International Scholarship Foundation and SRF (to H.O.). H.-W.C. is supported by a JSPS Postdoctoral Fellowship Program for Foreign Researchers and a Takeda Science Foundation research fellowship.

## Author contributions

H.O. conceived the project, H.-W.C. and H.O. designed the research. H.W.C. conducted most of the biochemical and histological experiments in collaboration with M.D., J.-M.F., S.H., H.C., H.H., R.T., M.S., N.M., Y.M., Y.Y., J.-i.Y., M.S., Mi.M., and S.H. H.-W.C., M. D., J.-M.F., K.M., Ma.M., and Mi.M. performed data analyses. H.-W.C. and H.O. drafted the paper supported by M.D., J.M.F. and S.H.

## Additional information

**Competing interests:** The authors declare no competing financial interests.

