## [Peer Review File · Nature Communications]

Reviewers' comments:

Reviewer #1 (Remarks to the Author):

To say it upfront, this a beautiful and well-written paper, summarizing an impressive body of technically impeccable data that convincingly support the authors' claims. The authors should be congratulated to this original and well conducted study.

Unlike most other tissues, the liver contains a large fraction of polyploid cells, and this fraction increases with age and/or under certain pathological conditions. In their study, the authors show that mice with null alleles for the circadian clock genes *Per1* and *Per2* contain a larger fraction of polyploid hepatocytes than wild-type mice. Using cultured primary hepatocytes from these mice they then demonstrate that these cells display abscission failure, probably because the midbody does not contain a sufficiently high activity of ERK1/2 kinase. ERK1/2 activity is upregulated through phosphorylation of ERK1 mostly by MEK1/2. Indeed, pERK1/2 immunohistochemistry indicated low staining for this phosphoprotein in the midbody. This could either be due to a lower phosphorylation or a higher dephosphorylation of ERK in *Per1/2* deficient cells. The authors show that ERK1/2 phosphatase (MKP1) rather than kinase (MEK1/2) dysregulation accounts for the observed deficiency in pERK1/2. Thus, MKP1 expression is constitutively upregulated in *Per1/2* knockout mice and diurnal in wild-type mice, perhaps because *Mkp1* transcription is under the control of the CLOCK:BMAL1 heterodimer, whose transactivation activity is annulled by PER-CRY complexes. Pharmacological approaches support the authors' conclusions in a compelling fashion.

I enthusiastically recommend publication of this nice piece and only have two relatively minor queries.

(1) Like in most studies on loss-of-function mutants of circadian clock genes, the term "circadian clock" is confounded with "circadian clock genes". Otherwise put, it is likely that constantly intermediate levels of PER and MKP1 would do just as well in preventing defects in abscission as cycling levels. This is actually supported by the pharmacological rescue experiments with hepatocytes from *Per1/2* knockout mice (page 9; Figure 4D). Clearly, the difference between "clock" and "clock genes" must be addressed and appropriate changes made in the text.

(2) Genome-wide ChIP-seq data for CLOCK-BMAL1 (Koike et al., 2012; Rey et al., 2011) and PER1/2 (Koike et al., 2012) are available. The authors should examine whether *Mkp1* is found among the genes bound by these transcriptional regulatory proteins. ChIP-seq experiments are generally more reliable than ChIP experiments in which the enrichment in the precipitated DNA is determined by q-PCR experiments (not RT-qPCR, as erroneously written on page 17 in Methods) such as the one shown in Figure 4C. In ChIP-seq data, the background DNA fragments caused by unspecific binding and contamination due to incomplete specificity in the immunoprecipitation procedure are randomly spread out over the entire genome (i.e. three billion base pairs), and specifically bound regions stick out as peaks composed of hundreds of sequence reads. It should be kept in mind that at any given moment in any given cell most TFs are bound to nonspecific, rather than specific sites. Moreover, in ChIP-seq assays the number of sequence reads in peaks can be compared between genes known to bind TFs based on genetic experiments (e.g. *Per1*, *Per2*, *Rev-erba*, *Dbp*). I suggest that gene browser data on *Mkp1* and the mentioned control genes from the Naef and Takahashi ChIP-seq papers are shown in a Supplemental Figure, obviously by citing these publications.

Koike, N., Yoo, S.H., Huang, H.C., Kumar, V., Lee, C., Kim, T.K., and Takahashi, J.S. (2012). Transcriptional architecture and chromatin landscape of the core circadian clock in mammals. *Science* 338, 349-354.

Rey, G., Cesbron, F., Rougemont, J., Reinke, H., Brunner, M., and Naef, F. (2011). Genome-wide and phase-specific DNA-binding rhythms of BMAL1 control circadian output functions in mouse

liver. PLoS biology 9, e1000595.

Reviewer #2 (Remarks to the Author):

Chao et al. generate and characterize a mouse model for specific deletion of Period gene homologues (Per1, Per2 and Per3). Per-null livers have increased hepatic polyploidy (enlarged hepatocytes with giant nucleus) with a specific localization around the central vein. The authors show in primary hepatocyte cultures that loss of Per genes promotes during mitosis abscission failure. Molecular characterization revealed that the Per-null hepatocytes present high levels of the mitogen-activated kinase phosphatase 1 (Mkp1). Increased Mkp1 activity inactivates the extracellular signal-regulated kinase (Erk1/2) and thus regulates abscission failure leading to polyploidization. Overall, the manuscript is well written, and experiments are well executed. However, I am concerned about interpretation of some of the data, and my concerns are outlined as Major concerns below.

MAJOR CONCERNS

1. Results on zone-specific polyploidization in Per-null mouse livers are not convincing. Polyploid hepatocytes in Per null livers seem to be more in mid lobule zone than in centrolobular zone (Figure 1a, c, d, h). There is a traditional 3-zone classification scheme in which zone 1 denotes the periportal zone and zone 3 denotes the centrolobular zone. Interestingly, recent papers identified enrichment of polyploid hepatocytes in the mid-lobule zone in mice livers (Morales-Navarrete et al., 2015 ; Tanami et al., 2016). Spatial distribution of ploidy classes along the liver lobule radial axis has to be re-analyzed with specific staining of periportal and pericentral area.

2. Post natal liver development is characterized by hepatocyte proliferation. It is established that there is a correlation between rate of proliferation and liver polyploidization (Gupta et al., 2000 ; Pandit et al., 2014 ; Hsu et al., 2016) during development. The enrichment of polyploid hepatocytes in Per-null livers is likely a consequence of enhanced proliferation during liver development. In fact supplementary Figure 5 presents data showing a higher labeling of KI67 in Per-null livers than control livers. The authors have to demonstrate during liver development (kinetics) the rate of hepatocyte proliferation (correlation with the genesis of polyploid hepatocytes).

3. In the liver, polyploidy is determined by the number of nuclei per cell and the DNA content of each nucleus. Polyploid hepatocytes have been demonstrated to be mainly generated following cytokinesis failure leading to the genesis of binucleate polyploid hepatocytes (Margall et al., 2007 ; Pandit et al., 2012 ; Hsu et al., 2016). Endoreplication is an alternative mechanism that leads to the genesis of mononucleate polyploid hepatocytes (Gentric et al., 2015 ; Panattoni et al., 2014 ; Diril et al., 2012). Per-null livers present (as described by the authors) enlarged hepatocytes with giant nucleus (polyploid mononucleate hepatocytes), suggesting that endoreplication preferentially occurs during liver development. Strangely, in vitro experiments on primary hepatocytes cultures demonstrated that Per-null hepatocytes with a reduced activity of Erk1/2 failed cytokinesis leading to the genesis of polyploid binucleate hepatocytes. The authors observed in culture that binucleate hepatocytes starting a new cell cycle performed successfully cytokinesis and generated polyploid mononucleate hepatocytes. First, the authors have to demonstrate why during the second cell cycle cytokinesis occurs although Erk is still down regulated. Moreover, they have also to demonstrate that the main mechanism inducing hepatocytes polyploidization in vivo is cytokinesis failure. In vivo cell cycle analysis can be performed after 2/3 hepatectomy by analyzing liver regeneration process in Per-null mice.

4. Per-null mice used in this study are total KO mice. It will be interesting to define if regulation of hepatic polyploidization by Per genes appeared to occur in a tissue-specific manner. What is the impact of depletion in primary mouse embryonic fibroblasts (MEF) ?

MINOR CONCERNS

1. In the introduction part, the authors claim : « Why and how hepatocytes develop polyploidy have not yet been addressed ». Since few years different factors have been shown to regulate hepatic polyploidy during liver development : E2F1/7/8, insulin, miR-122...
2. Figure 2 : the cell (cartoon) at T0 seems to be in S phase because at T40 minutes it is in anaphase. In this case, the cell at T0 is 8n.

Reviewer #3 (Remarks to the Author):

This is an interesting paper that clearly documents role for clock genes in pericentral mouse hepatocytes. The authors convincingly demonstrate that mice triply mutant in Per1, Per2 and Per3 have increased polyploidization in hepatocytes. They also use genetic and pharmacologic gain and loss of function studies to demonstrate that circadian oscillation of Mkp1 is a major mediator of this phenomenon. The data are very clear, the methodology is well controlled and the conclusions solid. I really have only minor suggestions for improvement...

- 1) The authors appear to have accepted the Axin2 pericentral hepatocytes as true stem cells. Currently, there is only the original Nature paper from the Nusse lab to support this idea. Many older papers strongly argue against streaming during normal liver homeostasis, both portal to central or central to portal. It is premature to talk about the diploid pericentral hepatocyte population as a stem cell and I would advise the authors to not emphasize this concept. In any case, this is irrelevant to their work.
- 2) Mitosis of polyploid hepatocytes has been tied to aneuploidy. In addition, perturbations of circadian rhythm have been tied to cancer. Have you karyotyped your Per-null hepatocytes? What is the incidence of HCC in your mutants?
- 3) Triple clock mutants don't exist in nature and even in these very unusual mice, cytokinesis failure occurs only in 20% of hepatocytes. Thus the connection between the clock and cytokinesis is subtle. This should be made very clear in the discussion.
- 4) With all due respect, references 13 and 14 don't really make any sense. Given the very infrequent rate of cell division in hepatocytes (once a year?), the finding of different percentages of binucleated cells at different times of day makes no sense at all. I would take these very old data with a grain of salt unless you have new data of your own.

Point-by-point reply to Reviewers:

1. Reviewer #1:

To say it upfront, this a beautiful and well-written paper, summarizing an impressive body of technically impeccable data that convincingly support the authors' claims. The authors should be congratulated to this original and well conducted study.

.....
I enthusiastically recommend publication of this nice piece and only have two relatively minor queries.

Reply: We very much appreciate this reviewer's evaluation of our work. The upfront remarks by this reviewer strengthen our motivation to complete this study. Scientific concerns from this reviewer are addressed below.

(1) Like in most studies on loss-of-function mutants of circadian clock genes, the term "circadian clock" is confounded with "circadian clock genes". Otherwise put, it is likely that constantly intermediate levels of PER and MKP1 would do just as well in preventing defects in abscission as cycling levels. This is actually supported by the pharmacological rescue experiments with hepatocytes from Per1/2 knockout mice (page 9; Figure 4D). Clearly, the difference between "clock" and "clock genes" must be addressed and appropriate changes made in the text.

Reply: We accept this proposal. We completely agree with the referee, and changed the four parts of *circadian clock* in the text, as follows,

1. (Summary, line 11) These data provide clear evidence that the **circadian Period genes** not only orchestrate dynamic changes in metabolic activity, but also regulate homeostatic self-renewal of hepatocytes through Mkp1-Erk1/2 signaling pathway.
2. (Introduction, last sentence) These findings indicate that the **circadian clock genes** not only orchestrate daily metabolic changes in hepatocytes, but also regulate homeostatic hepatocyte self-renewal through Mkp1-pErk1/2 signaling pathway.
3. (Result, Constant upregulation of Mkp1 in Per-null mice: last sentence) Based on the observed negative correlations between Mkp1 and pErk1/2 abundance in WT and Per-null mice (**Fig. 4b**), published genome-wide ChIP-seq data for Clock and Bmal1^{39,40} (**Supplementary Fig. 9**), and a chromatin immunoprecipitation assay that demonstrated the circadian time-specific recruitment

of *Per1* to the E-box enhancer element in the *Mkp1* promoter (**Fig. 4c**), we propose that the **clock gene**-controlled protein *Mkp1* regulates phosphorylation of Erk1/2 in the liver.

4. (Discussion, 2nd paragraph, last sentence) These findings open a new dimension of **clock gene function** in cell biology: intracellular **clock components** directly regulate cytokinesis, and *Periods* ablation results in abscission failure through circadian dysregulation of *Mkp1*.

As shown above, we exchange “circadian clock” to “circadian *Period* genes” in (1), to “circadian clock genes” in (2), and to “clock gene-” in (3). In (4), we changed from “clock function” and “circadian clockwork” to “clock gene function” and “clock components”, respectively.

(2) Genome-wide ChIP-seq data for CLOCK-BMAL1 (Koike et al., 2012; Reyt et al., 2011) and PER1/2 (Koike et al., 2012) are available. The authors should examine whether Mkp1 is found among the genes bound by these transcriptional regulatory proteins. ChIP-seq experiments are generally more reliable than ChIP experiments in which the enrichment in the precipitated DNA is determined by q-PCR experiments (not RT-qPCR, as erroneously written on page 17 in Methods) such as the one shown in Figure 4C. In ChIP-seq data, the background DNA fragments caused by unspecific binding and contamination due to incomplete specificity in the immunoprecipitation procedure are randomly spread out over the entire genome (i.e. three billion base pairs), and specifically bound regions stick out as peaks composed of hundreds of sequence reads. It should be kept in mind that at any given moment in any given cell most TFs are bound to nonspecific, rather than specific sites. Moreover, in ChIP-seq assays the number of sequence reads in peaks can be compared between genes known to bind TFs based on genetic experiments (e.g. Per1, Per2, Rev-erba, Dbp). I suggest that gene browser data on Mkp1 and the mentioned control genes from the Naef and Takahashi ChIP-seq papers are shown in a Supplemental Figure, obviously by citing these publications.

Koike, N., Yoo, S.H., Huang, H.C., Kumar, V., Lee, C., Kim, T.K., and Takahashi, J.S. (2012). Transcriptional architecture and chromatin landscape of the core circadian clock in mammals. Science 338, 349-354.

Rey, G., Cesbron, F., Rougemont, J., Reinke, H., Brunner, M., and Naef, F. (2011). Genome-wide and phase-specific DNA-binding rhythms of BMAL1 control circadian output functions in mouse liver. PLoS biology 9, e1000595.

Reply: We accept this proposal. As this reviewer suggested, we examined published ChIP-seq data and found that *Mkp1* promoter is bound by CLOCK (Yoshitane *et al.*, 2014) and BMAL1 (Annayev *et*

al., 2014). Genome browser pictures constructed from these studies are now shown in **Supplementary Fig 9** (and also below for the reviewer).

Supplementary Fig. 9

Figure legend of Supplementary Fig. 9

Visualization of previously reported ChIP-seq signals for BMAL1 (blue) and CLOCK (green) on *Mkp1* (*Dusp1*) and *Per2*.

Blue: BMAL1 ChIP-seq data reported by Annayev *et al.* in J Biol Chem 289, 5013-24, 2014 (SRA ID: SRX45001 and SRX45002).

Green: CLOCK ChIP-seq data reported by Yoshitane *et al.* in Mol Cell Biol 34, 1776-87, 2014 (SRA ID: DRX011960).

Arrow indicates the position of the *Mkp1* E-box, to which the binding of CLOCK and BMAL1 has been confirmed in ChIP-seq experiments, albeit with lower affinity than to the *Per2* E'-box (asterisk).

In both reports, livers at ZT8 were analyzed, and peak calling analysis revealed significant peaks in the proximity of this E-box.

BigWig files that correspond to each ChIP-seq experiment were downloaded from the ChIP-Atlas website (<http://chip-atlas.org/>) and visualized with the Integrative Genomics Viewer (<http://software.broadinstitute.org/software/igv/>).

We also checked ChIP-seq data from Rey *et al.* (2011) and Koike *et al.* (2012), suggested by the reviewer, but no significant peaks were detected in this region. These observations imply that the binding of clock proteins to the *Mkp1* E-box is affected by the experimental conditions used. In this regard, it is interesting to note that the PER1 antibody that we used for our present ChIP-qPCR experiment (**Fig. 4c**) differs from the one Koike *et al.* used.

2. Reviewer #2

Overall, the manuscript is well written, and experiments are well executed. However, I am concerned about interpretation of some of the data, and my concerns are outlined as Major concerns below.

Reply: We very much appreciate this reviewer's valuable comments and positive evaluation of our work. He/she pointed out important points that we addressed as follows.

1. Results on zone-specific polyploidization in Per-null mouse livers are not convincing. Polyploid hepatocytes in Per null livers seem to be more in mid lobule zone than in centrolobular zone (Figure 1a, c, d, h). There is a traditional 3-zone classification scheme in which zone 1 denotes the periportal zone and zone 3 denotes the centrolobular zone. Interestingly, recent papers identified enrichment of polyploid hepatocytes in the mid-lobule zone in mice livers (Morales-Navarrete et al., 2015 ; Tanami et al., 2016). Spatial distribution of ploidy classes along the liver lobule radial axis has to be re-analyzed with specific staining of periportal and pericentral area.

Reply: We decided to perform this study from our original finding of histological analysis using Hematoxylin-Eosin staining. From this analysis, shown typically in **Fig. 1c**, it is very clear that hepatocytes are divided into two separable groups: namely, a group surrounding the central vein and another group surrounding the portal vessels.

Fig.1c

Nevertheless, this reviewer's comment is important, as prior conventional histological studies utilizes three zone classification of the lobule: 3 equally divided from the distance of PV to CV, defining the centrilobular zone, midlobular zone, and periportal zone. Following the indicated newly published references by the referee (Morales-Navarrete H et al. *Elife*. 4. pii: e11214, 2015; Tanami S et al. *Cell Tissue Res*. 20126), we analyzes in more detail along the CV-PV axis.

We subdivide the CV-PV axis into 15 parts, the lower number indicating the region closer to the central vein, as shown in **Fig.1d** and **Supplementary Fig.3**. In this analysis, regions 1-5 correspond to the centrilobular zone, 6-10 to midlobular zone, and 11-15 to periportal zone in the traditional 3-zone classification. We analyzed five mouse livers in each part.

Fig. 1d

Figure legend of Fig.1d

(d) Zone-specific nuclear sizes stained with Hoechst-33342 of WT and *Per*-null mice. The bar graph shows the distribution of variant hepatic nuclear size along the CV-PV axis. The CV-PV axis was subdivided into 15 parts (left schema), the lower number indicating the region closer to central vein region. Regions 1-5 correspond to the centrilobular zone, 6-10 to midlobular zone, and 11-15 to periportal zone in the traditional 3-zone classification. Note significant increase of nuclear diameter in *Per*-null hepatocytes compared to WT was observed in areas from 3 to 8, across centrilobular to midlobular zones, and thus, we describe these cells as centro-midlobular hepatocytes (CMH) separating from periportal hepatocytes (PH). The results were generated from five mouse livers for each group, and three different lobules were analyzed in each liver. Values represent the mean \pm SEM., * $P < 0.05$, ** $P < 0.01$, *** $P < 0.001$.

These data clearly demonstrate that *Per*-null hepatocytes display larger nuclei (compared to WT) from part 3 to 8 subdivisions, which correspond to outer centrilobular zone to inner midlobular zone. The

data also indicate that hepatocytes in the periportal lobule exhibit similar nuclear size in *Per*-null and WT liver.

Although not shown in this paper, here we attached the calculated value of the conventional three zone classification, from Fig.1d, here for the interest of the referee. As shown clearly, we found that centrilobular hepatocytes display no significant difference in nuclear size with midlobular hepatocytes, but significantly larger nuclei of periportal hepatocytes (see the attached figure of left side).

From the above analysis, we decided to use a more accurate histological description of polyploid hepatocytes localization: these polyploid cells are now designed as to centro-midlobular hepatocytes (CMH), distinct from periportal hepatocytes (PH).

We thank the reviewer very much for consolidating our findings to show us the recent histological studies for zonation of liver lobules.

2. Post natal liver development is characterized by hepatocyte proliferation. It is established that there is a correlation between rate of proliferation and liver polyploidization (Gupta et al., 2000 ; Pandit et al., 2014 ; Hsu et al., 2016) during development. The enrichment of polyploid hepatocytes in Per-null livers is likely a consequence of enhanced proliferation during liver development. In fact supplementary Figure 5 presents data showing a higher labeling of KI67 in Per-null livers than control livers. The authors have to demonstrate during liver development (kinetics) the rate of hepatocyte proliferation (correlation with the genesis of polyploid hepatocytes).

Reply: To answer this question, we first try to analyze the developmental changes in the distribution of Ki67-positive cells, but in adult liver lobules the amount of Ki67-positive cells was extremely low and highly variable, between animals and between lobules of the same animals, making this analysis unreliable.

As an alternative to address this comment, we instead performed BrdU-incorporation analysis at 2, 4, 6, 8 and 10 weeks of age in both WT and *Per*-null mice (each, n=5). BrdU was injected once (0.15 mg/g body weight, intraperitoneally) and 1 hour later animals were sacrificed. Compared to the previous Ki67 method which labels cells out of the G₀ phase, BrdU method directly measures cells at S-phase in cell cycle, and has been used successfully by many laboratories. So, we now include this

data in Supplementary figure 7i. In order to show the correlation suggested by the reviewer, we also quantified the nuclear sizes of hepatocytes at each developmental stage (n=5 at each stage) (Supplementary Fig. 7 h, i.)

Supplementary Fig. 7h,i

Figure legend of Supplementary Fig. 7h,i and Supplementary discussion of Supplementary Fig. 7

(h, i) Quantitative analyses of the development of nuclear sizes (i) and the number of BrdU-incorporated cells (j) in WT and *Per*-null liver (mean \pm SEM; n=5 at each developmental stage at each genotype; injection protocol and tissue preparation are the same of (e)). (i) In both genotypes, the average nuclear size increases according to the development from 2 to 10 weeks of postnatal days. Although there were no genotype difference at 2 weeks old mice (see also Fig. 1g), increase of genotype-specific nuclear ploidy became prominent at *Per*-null mice at 4 weeks old, and it increases in proportion to gain age until 10 weeks. In the two-way ANOVA, the main effect of time and genotype were both significant. (j) Developmental analysis of BrdU demonstrates a sharp age-dependent decline BrdU incorporation in both genotypes: the actual labelling of hepatocytes decreases more than 50 times at 10 weeks compared to 2 weeks. In two-way ANOVA, only the main effect of time was significant, while genotype and interactions effects were not significant in both CMH and PH. These data clearly show that age-dependent cell-size increase (showing the degree and the proportion of polyploidy cells) is inversely proportional to the number of proliferating cells, which decreases with age.

We also discussed this finding in the following **Supplementary Discussion for Supplementary Fig. 7.**

Supplementary Discussion for Supplementary Fig. 7: Difference of pErk1/2 expression in pluripotent pericentral stem cells (PCS) and mature hepatocytes surrounding PCS. It is interesting to note that the *Mkp1*-pErk1/2 pathway appears weak in pericentral stem cells (PCS) (Wang et al. 2015) lining the central vein, but more active in mature CMH surrounding PCS, which are possible descendants of PCS

(Wang et al. 2015). Since Ki67-expressing cells, representing the cells in growth fraction (cells out of G_0), were sparsely distributed but dominantly located in the midlobular region (**Supplementary Fig. 7g**), the mature CMH thus appear to play a central role in homeostatic renewing process of organ mass, in contrast to pluripotent PCS stem cells. The separation of pluripotent stem cells and proliferative cells is commonly known in most tissues such as small intestine¹⁴. Here, proliferative hepatocytes were dispersed among silent G_0 hepatocytes (**Supplementary Fig. 7g**). Although a correlation between rate of proliferation and liver polyploidization has been reported (Gupta et al., 2000; Pandit et al., 2014; Hsu et al., 2016), no such correlation was found in Per-null liver (**Supplementary Fig. 7i**).

Above is the answer to the reviewer's question whether there is a correlation between rate of proliferation and liver polyploidization. We decided to keep the qualitative data of Ki67 labeling in **Supplementary Fig.7g**. but replaced the previous incomplete analysis of Ki67 (previous Sup. Fig.5h) with a more quantitative analysis of proliferation, shown in the new **Supplementary Fig. 7i**. Although a correlation between rate of proliferation and liver polyploidization has been reported (Gupta et al., 2000; Pandit et al., 2012; Hsu et al., 2016) as referee indicated, no such correlation was found in Per-null liver.

3. In the liver, polyploidy is determined by the number of nuclei per cell and the DNA content of each nucleus. Polyploid hepatocytes have been demonstrated to be mainly generated following cytokinesis failure leading to the genesis of binucleate polyploid hepatocytes (Margall et al., 2007 ; Pandit et al., 2012 ; Hsu et al., 2016). Endoreplication is an alternative mechanism that leads to the genesis of mononucleate polyploid hepatocytes (Gentric et al., 2015 ; Panattoni et al., 2014 ; Diril et al., 2012). Per-null livers present (as described by the authors) enlarged hepatocytes with giant nucleus (polyploid mononucleate hepatocytes), suggesting that endoreplication preferentially occurs during liver development. Strangely, in vitro experiments on primary hepatocytes cultures demonstrated that Per-null hepatocytes with a reduced activity of Erk1/2 failed cytokinesis leading to the genesis of polyploid binucleate hepatocytes. The authors observed in culture that binucleate hepatocytes starting a new cell cycle performed successfully cytokinesis and generated polyploid mononucleate hepatocytes. First, the authors have to demonstrate why during the second cell cycle cytokinesis occurs although Erk is still down regulated. Moreover, they have also to demonstrate that the main mechanism inducing hepatocytes polyploidization in vivo is cytokinesis failure. In vivo cell cycle analysis can be performed after 2/3 hepatectomy by analyzing liver regeneration process in Per-null mice.

Reply: The question that why cells in the first mitosis failed to divide, but the second mitosis succeed is a difficult question which has not been addressed. In the developmental study, liver polyploidy formation occurs in two steps: first the binuclear formation ($2n \times 2$) with abortive cell cycle, and the second, the genesis of two mononuclear tetraploid ($4n \times 1 \times 2$) hepatocytes (Guidotti et al., J Biol Chem 278:19095-19101, 2003). Thus, for the explanation of the polyploidy, it is important to address these two steps of cell cycle.

We presented the evidence that in the first step, that is abortive cell cycle, clock-Mkp1-Erk plays a key for the genesis of binuclear formation. However, we do not think this pathway is important for the second step for the generation of mononuclear tetraploid cells, because we observed complete cytokinesis when binuclear cell divides even in pErk-down regulated *Per*-null mice.

Important contributions were presented by the group of Dr. Desdouets for the mechanism of the second step, when they demonstrated that the two paired centrosomes contractile might be the cause (Guidotti et al., J Biol Chem 278:19095-19101, 2003). In hepatocyte primary culture, they demonstrated that the binuclear cells exhibited two centrosomes in G1 that were duplicated during S phase and then 4 centrosomes clustered by pairs at opposite poles of the cell during metaphase. This event led only to mononuclear $4n$ progeny and maintained the tetraploidy status of hepatocytes. So, we think this centrosome formation, not pErk, will play a critical role of the second step of the hepatocyte polyploidy formation.

The reviewer asks to us whether pErk is down regulated or not, in the second step of mitosis. However, it is very difficult to detect the timing of the second mitosis or the first mitosis since so far we are limited to the use of immunohistochemistry to detect pErk signaling. If we could perform pErk labelling *in vivo* and in real time by using next generation FRET (Aoki et al. Mol Cell. 52:529-540, 2013), it would provide good insights.

So far, we have checked the localization of pErk in all steps of mitosis and cytokinesis, and found that pErk is down regulated in all *Per*-null hepatocytes. After checking hundreds of photos of pErk staining in cells of abscission step, we never found clear pErk staining in the midbody in *Per*-null mice.

For the curiosity of the reviewer, we attached pErk staining in all steps of mitosis. In WT hepatocytes, we found that pErk1/2 was strongly expressed at the midbody during the abscission stage, in addition to perinuclear structures in all step. Notably, in *Per*-null hepatocytes, perinuclear pErk staining tended to diminishes in whole step of mitosis, but the most prominent finding is the absence of midbody localization of pErk1/2 in all cases.

pErk1/2 expression in the course of mitosis in cultured hepatocytes.

In all stages of mitosis, pErk1/2 was expressed in perinuclear cytoplasm in both WT and *Per*-null hepatocytes. In abscission stage, pErk1/2 was expressed strongly on the midbody in addition to the perinuclear staining. There was a tendency that pErk1/2 signal is decreased in *Per*-null cells compared to those in WT. Among them, pErk1/2 on the midbody was most strongly affected with complete abolishment in *Per*-null hepatocytes during abscission (insets). Tubulin (green) outlines spindle fibers and the midbody during mitosis. Scale bars, 20 μ m.

Concluding all, among two steps of liver polyploidy, we think that pErk will work in the first step for the genesis of binuclear cell formation, and for the second, the genesis of mononuclear tetraploid, other mechanisms such as paired centrosomes might have the pivotal role.

4. Per-null mice used in this study are total KO mice. It will be interesting to define if regulation of hepatic polyploidization by Per genes appeared to occur in a tissue-specific manner. What is the impact of depletion in primary mouse embryonic fibroblasts (MEF) ?

Thank you for the important question. Since *Per* genes are expressed virtually in all cells of the body, we examined the polyploidy in all major tissues of the body. We performed a systemic histological survey of tissue morphology by comparing a series of conventional hematoxylin and eosin (HE)-stained sections from brain, lung, heart, kidney, liver, intestine, colon, pancreas, spleen, adrenal gland, testis, and skin samples from *Per*-null mice with those from wild type (WT) mice. Among all tissues examined, a marked difference between genotypes was found only in the morphology of the liver with the appearance of macronucleated hepatocytes in *Per*-null liver

From these findings, we speculate the present Per-MKP1-pErk pathway may be only crucial for division of adult hepatocytes. This does not deny the possibility that this pathway work in the normal division of cells in other cells. Perhaps, it is possible that this pathway is crucial for pathological conditions, which study should be done in future studies. In this context, MEF cells are very interesting, and we are currently trying to use them in future studies.

We attached this finding in new **Supplementary Fig. 2**.

Supplementary Fig. 2

Figure legend of Supplementary Fig. 2.

Histological sections in various organs in WT and *Per*-null mice.

We performed hematoxylin and eosin staining in (a) small intestine, (b) liver, (c) large intestine (colon), (d) pancreas (exocrine gland and islet of Langerhans), (e) kidney (cortex), (f) adrenal (cortex), (g) lung (alveolus and bronchial epithelium), (h) testis, (i) skin (epithelium and hair follicle), (j) spleen (white pulp and red pulp), (k) cardiac muscle, and (l) cerebral cortex in WT and *Per*-null mice. Paraformaldehyde-fixed paraffin embedded sections (5 μ m in thickness) were histologically analyzed by light-microscopy (Olympus). Note no polyploidy was observed in all tissue examined except liver. Scale bars, 100 μ m.

MINOR CONCERNS In

- 1. In the introduction part, the authors clame : « Why and how hepatocytes develop polyploidy have not yet been adressed ». Since few years different factors have been shown to regulate hepatic polyploidy during liver developement : E2F1/7/8, insulin, miR-122...*
- 2. Figure 2 : the cell (cartoon) at T0 seems to be in S phase because at T40 minutes it is in anaphase. In this case, the cell at T0 is 8n.*

Thank you for these indications, we added the references as indicated. Also, we corrected 4n to 8n.
Thank you very much.

3. **Reviewer #3**

This is an interesting paper that clearly documents role for clock genes in pericentral mouse hepatocytes. The authors convincingly demonstrate that mice triply mutant in Per1, Per2 and Per3 have increased polyploidization in hepatocytes. They also use genetic and pharmacologic gain and loss of function studies to demonstrate that circadian oscillation of Mkp1 is a major mediator of this phenomenon. The data are very clear, the methodology is well controlled and the conclusions solid. I really have only minor suggestions for improvement...

Reply: We very much appreciate this reviewer's valuable comments and positive evaluation of our work. All comments were very valuable, leading us to change the text as follows.

1) The authors appear to have accepted the Axin2 pericentral hepatocytes as true stem cells. Currently, there is only the original Nature paper from the Nusse lab to support this idea. Many older papers strongly argue against streaming during normal liver homeostasis, both portal to central or central to portal. It is premature to talk about the diploid pericentral hepatocyte population as a stem cell and I would advise the authors to not emphasize this concept. In any case, this is irrelevant to their work.

Thank you for your advice of the present status of Axin2-Nusse theory. We accept the reviewer's proposal and rewrite the sentences in INTRODUCTION and DISCUSSION to clarify that self-renewal is considered to occur from three possible origins including self-divisions of mature hepatocytes, Lgr5 stem cells in periportal area, and Axin2-centrilobular area.

2) Mitosis of polyploid hepatocytes has been tied to aneuploidy. In addition, perturbations of circadian rhythm have been tied to cancer. Have you karyotyped your Per-null hepatocytes? What is the incidence of HCC in your mutants?

Reply: Thank you for the comments. Instead of chromosome karyotyping of individual hepatocytes, we analyzed DNA ploidy including diploidy, polyploidy, and aneuploidy of hepatocytes of WT and *Per*-null mice at 3 and 12 weeks by DNA flow cytometry. Although polyploidy has been tied to aneuploidy, which arises through mitotic defects, chromosome missegregation, or chromosome catastrophe, and may give rise to hepatocellular carcinoma, we found no obvious aneuploid “population” either in WT or in *Per*-null hepatocytes in the current study as shown in Figure 1h.

Dysregulation of circadian rhythm is likely to accelerate malignant transformation of various cells as reported previously in malignant lymphoma (Fu et al., *Cell* 111:41-50, 2002) and breast cancer (Climent et al., *J Clin Oncol* 28:3770-3778, 2010). Furthermore, Fu and colleagues demonstrated that chronic jet-lag increased incidence of hepatocellular carcinoma (HCC) in a recent paper (Kettner et al., *Cancer Cell* 30:1-16, 2016). However, according to their report, HCC was observed at about 19-22 months old mice. Even the youngest mouse having HCC was 10 months old. In our current study, we checked *Per*-null mice until 6 months of age to see if HCC occurs in their liver, but none of the *Per*-null mice harbors HCC. Our observations probably indicate that additional genetic events and/or aging periods are required for carcinogenesis in the liver.

Since we think that this discussion is important, we incorporate this discussion in the last 2nd paragraph in DISCUSSION of the main text, as follows.

Dysregulation of circadian rhythm is likely to accelerate malignant transformation of various cells as reported previously in malignant lymphoma⁵³ and breast cancer⁵⁵. Of particular interest is the fact that mitosis of polyploid hepatocytes has been tied to aneuploidy^{16,56}, which arises through mitotic defects, chromosome missegregation, or chromosome catastrophe, and may give rise to hepatocellular carcinoma. However, we found no obvious aneuploid “population” either in WT or in *Per*-null hepatocytes in our flow cytometric analysis as shown in **Figure 1h**. Furthermore, we found none of the *Per*-null mice harbors hepatocellular carcinoma (HCC) until 6 months of age. Fu and colleagues recently demonstrated that chronic jet-lag increased incidence of hepatocellular carcinoma (HCC) in a recent paper⁵⁷ in both wild type and clock gene-disrupted mice. However, according to their report, HCC was observed at aged mice: even the youngest mouse bearing HCC was 10 months old. Thus, our observations in *Per*-null mice probably indicate that additional

environmental events and/or aging, accompanied by genetic/epigenetic alterations, are required for carcinogenesis in the liver.

3) Triple clock mutants don't exist in nature and even in these very unusual mice, cytokinesis failure occurs only in 20% of hepatocytes. Thus the connection between the clock and cytokinesis is subtle. This should be made very clear in the discussion.

We added one sentence for the role clock on cytokinesis in DISCUSSION (the last paragraph).

However, it should be kept in mind that cytokinesis is a complex phenomenon: in the abscission stage alone, dozens of regulators were identified⁴⁹, the *Per*-Mkp1-pErk pathway described here being only one among many causes of cytokinesis failure.

4) With all due respect, references 13 and 14 don't really make any sense. Given the very infrequent rate of cell division in hepatocytes (once a year?), the finding of different percentages of binucleated cells at different times of day makes no sense at all. I would take these very old data with a grain of salt unless you have new data of your own.

Thank you for your advice. I erased these citations in the text.

Reviewers' comments:

Reviewer #1 (Remarks to the Author):

As mentioned in my report for the original submission, I find this to be a beautiful paper. The authors addressed my minor suggestions in a satisfactory way in the revised version and the rebuttal letter. In the latter, they point out that PER and CLOCK:BMAL1 binding sites within the Mkp1 promoter can be found in some, but not all published ChIP-seq experiments. Since PER proteins are corepressors and thus are bound only indirectly to their cognate DNA elements, the extent of cross-linking by formaldehyde may determine whether they provide signals in ChIP-seq experiments or not. Otherwise put, only if PERs are crosslinked to the relevant DNA-binding transcription factors (e.g. CLOCK:BMAL1), will they co-immunoprecipitate DNA. So the authors' argument in their rebuttal letter "These observations imply that the binding of clock proteins to the Mkp1 E-box is affected by the experimental conditions used" is plausible. In conclusion, I have no outstanding issues and recommend publication.

Reviewer #2 (Remarks to the Author):

In this new version, there is still a lack of data about molecular mechanisms that lead to polyploidization in vivo and there is still discrepancy between in vitro and in vivo data. As I already mentioned in my previous review there is in the liver mainly two types of polyploidization processes one that induces nuclear ploidy (genesis of mononuclear polyploid cells) and the other one that induces cellular ploidy (genesis of binuclear polyploid cells). The authors observed in the Per-null model in vivo alteration of nuclear ploidy (as it is described at least in Figure 1, e.g. line 109: this increase in nuclear size..., line 119 larger centro-mid lobular hepatocytes nuclei). By contrast, in vitro, on primary hepatocytes cultures, they demonstrated that genetic ablation of Periods genes triggers abscission failure: cellular ploidy (Figure 2-3-4, e.g. line 148 to form a binuclear cell). Previous publications have demonstrated that silencing of specific factors (miR122, E2F7/E2F8) that controlled cytokinesis events impacts only on cellular ploidy (same results in vitro and in vivo with a specific impact on binuclear fraction with no modification of nuclear size) (e.g. Hsu et al., Hepatology 2016; Pandit et al., Nature Cell Biology., 2012;). I have asked in my previous review complementary in vivo experiments to reinforce the paper. There is no new data on the process of polyploidization in vivo except Figure 7H. This figure illustrates that nuclear size increases during liver development reflecting more an endoreplication process. To explain the in vitro results, the authors argue that Desdouets and collaborators have previously demonstrated that division of binuclear hepatocytes lead to the genesis of mononuclear cells: « In hepatocyte primary culture, they demonstrated that the binuclear cells exhibited two centrosomes in G1 that were duplicated during S phase and then 4 centrosomes clustered by pairs at opposite poles of the cell during metaphase. This event led only to mononuclear 4n progeny and maintained the tetraploidy status of hepatocytes. ». Coming back into the JBC paper, Dr Desdouets demonstrated that clustering of centrosomes in binuclear hepatocytes is essential to maintain a polyploid content and to avoid the genesis of aneuploid contingent. In fact, the binuclear contingent can either perform a normal cell cycle (genesis of polyploid mononuclear fraction) but can also fail again cytokinesis (genesis of polyploid binuclear fraction). This mechanism has been described in different papers and reviews (e.g. Duncan AW, Semin Cell Dev Biol, 2013).

Point-by-point reply to Reviewers:

Reviewer #1 (Remarks to the Author):

As mentioned in my report for the original submission, I find this to be a beautiful paper. The authors addressed my minor suggestions in a satisfactory way in the revised version and the rebuttal letter. In the latter, they point out that PER and CLOCK:BMAL1 binding sites within the Mkp1 promoter can be found in some, but not all published ChIP-seq experiments. Since PER proteins are corepressors and thus are bound only indirectly to their cognate DNA elements, the extent of cross-linking by formaldehyde may determine whether they provide signals in ChIP-seq experiments or not. Otherwise put, only if PERs are crosslinked to the relevant DNA-binding transcription factors (e.g. CLOCK:BMAL1), will they co-immunoprecipitate DNA. So the authors's argument in their rebuttal letter "These observations imply that the binding of clock proteins to the Mkp1 E-box is affected by the experimental conditions used" is plausible.

In conclusion, I have no outstanding issues and recommend publication.

REPLY:

Thank you very much for appreciation comment and the support to our paper. As the referee pointed out the difference of the ChIP-seq data of published paper might reflect the evidence that PER proteins are corepressors. Thank you for your appreciation to our data.

Reviewer #2 (Remarks to the Author):

In this new version, there is still a lack of data about molecular mechanisms that lead to polyloidization in vivo and there is still discrepancy between in vitro and in vivo data. As I already mentioned in my previous review there is in the liver mainly two types of polyploidization processes one that induces nuclear ploidy (genesis of mononuclear polyploidy cells) and the other one that induces cellular ploidy (genesis of binuclear polyploid cells). The authors observed in the Per-null model in vivo alteration of nuclear ploidy (as it is described at least in Figure 1, e.g. line 109: this increase in nuclear size..., line 119 larger centro-mid lobular hepatocytes nuclei). By contrast, in vitro, on primary hepatocytes cultures, they demonstrated that genetic ablation of Periods genes triggers abscission failure: cellular ploidy (Figure 2-3-4, e.g. line 148 to form a binuclear cell).

REPLY:

As the reviewer #2 mentioned, cytokinesis failure can give rise to both mononuclear polyploidy cells and binuclear polyploid cells. In our original manuscript (particularly in **Fig. 1**), however, we simply focused only on the most conspicuous phenotype of polyploidy: the increase in nuclear size. This is not, as the reviewer misunderstood, a discrepancy with *in vitro* data since we have shown no data that binuclear polyploid cells do not occur *in vivo*. Nonetheless, the comments provided by the reviewer in the second revision led us to realize that our initial *in vivo* analyses, emphasizing the size of the nuclei regardless of their number per cell, could have been seen as incomplete. Since we were kindly given the opportunity to revise our

manuscript again, we here provide a complete set of new data on the number of binuclear hepatocytes during normal liver development in *Per*-null and wild-type mice.

The referee indicated an important notion that an increase in liver polyploidy can originate from the accumulation of mononuclear hepatocytes with polyploid nucleus (i.e. nuclear ploidy) and/or that of binuclear cells (cellular ploidy), so this time we precisely analyzed liver sections by double β -catenin/Hoechst-33342 immunohistochemistry, quantifying nuclear size and number per cell. When analyzing 10 weeks of mice, we observed that both cellular and nuclear polyploidization are accelerated in hepatocytes of *Per*-null centrilobular and midlobular zones, characterized by a highly significant increase in 8n binuclear (2 x 8n) cells as well as 16n mononuclear hepatocytes in the centrilobular zone (**Supplementary Fig. 3**).

Developmental analysis of the liver from 2 weeks to 8 weeks also supports that both cellular and nuclear polyploidy occurred during development. Indeed, both binuclear and mononuclear polyploid cells were observable at 4 weeks in mouse liver, with genotype-specific binucleation and polyploidization differences detected as early as 6 weeks that became highly significant at 8 weeks (**Supplementary Fig. 4**), suggesting their etiology is developmentally connected.

Thus, by this *in vivo* developmental analysis, we clearly demonstrated that both cellular and nuclear polyploidization were accelerated *in vivo* in *Per*-null liver, similar to what was observed in the video of *in vitro* primary culture of hepatocytes. During the recording, two cell cycles occurred, abscission in the first cell cycle failed and produced binuclear cells, whereas in the second cycle, a complete cytokinesis produced polyploid hepatocytes with a single enlarged polyploid nucleus (**Supplementary Fig. 6g, Supplementary Video 2**).

We added one paragraph describing the above findings in the manuscript (after the description of the analysis of flow cytometry), accompanying new **Supplementary Fig. 3** and **Supplementary Fig. 4**, as follows.

(Main Text)

“Since an increase in liver polyploidy can originate from the accumulation of mononuclear hepatocytes with polyploid nucleus (i.e. nuclear ploidy) and/or that of binuclear cells (cellular ploidy)^{33,34}, we sought to refine our analysis. Cytometrical analysis of liver sections by β -catenin/Hoechst-33342 immunohistochemistry, quantifying nuclear size and number per cell, revealed that both cellular and nuclear polyploidization are accelerated in *Per*-null CMH, characterized by a highly significant increase in 8n binuclear (2 x 8n) cells as well as 16n mononuclear hepatocytes in the centrilobular zone (**Supplementary Fig. 3**). The developmental onset of binucleation and polyploidization coincides, observable by 6 weeks of

age to become significant by 8 weeks (**Supplementary Fig. 4**), suggesting their etiology is developmentally connected.”

Supplementary Figure 3

Legend of Supplementary Fig. 3

Identification of binuclear cells by the double-stained liver sections with cell membrane marker β -Catenin and nuclear marker Hoechst-33342.

By β -Catenin immunohistochemistry, we analyzed cellular and nuclear ploidy in WT (**a, c**) and *Per*-null (**b, d**) mice (10 weeks). (**c**) and (**d**) are high power field photomicrographs of indicated

areas of (a) and (b), respectively. We divided the liver lobule to 15 as indicated lines, and counted the mononuclear 2n cells (2n x 1), binuclear 2n cells (2n x 2), mononuclear 4n cells (4n x 1), binuclear 4n cells (4n x 2), mononuclear 8n cells (8n x 1), binuclear 8n cells (8n x 2), mononuclear 16n cells (16n x 1), binuclear 16n cells (16n x 2), mononuclear 32n cells (32n x 1) and binuclear 32n cells (32n x 2) according to the nuclear size of 2n cells in each section. We show the incidence (%) of each ploidy state in mononucleated and binucleated cells within each zone (centrilobular, midlobular and periportal zone) of each hepatic lobe (n=360). Scale bars, 100 μ m in (a, b) and 20 μ m in (c, d).

Supplementary Fig.4

Legend of Supplementary Fig. 4

Developmental change of nuclear ploidy and binucleation detected by β -Catenin immunohistochemistry and nuclear staining of Hoechst-33342.

We analyzed developmental changes in nuclear ploidy by histochemistry using β -Catenin and Hoechst-33342. We analyzed the incidence (%) of each ploidy state of mononucleated (x1) and binucleated (x2) cells within each zone (centrilobular, midlobular and periportal zone) of each

hepatic lobes (n=980). Scale bars, 100 μ m. Note hepatocytes increase their polyploidization age-dependently, except those in periportal zone.

Previous publications have demonstrated that silencing of specific factors (miR122, E2F7/E2F8) that controlled cytokinesis events impacts only on cellular ploidy (same results in vitro and in vivo with a specific impact on binuclear fraction with no modification of nuclear size). I have asked in my previous review complementary in vivo experiments to reinforce the paper. There is no new data on the process of polyploidization in vivo except Figure 7H. This figure illustrates that nuclear size increases during liver development reflecting more an endoreplication process.

REPLY:

Silencing specific factors controlling the cytokinesis (miR122, E2F7/E2F8) demonstrated an increase in binucleation (e.g. Hsu et al., Hepatology 2016; Pandit et al., Nature Cell Biology., 2012) without nuclear polyploidization suggested that these factors might be involved only in binucleation. However, the new *in vivo* data we added clearly show the deletion of *Per* genes led to an increase of both cellular and nuclear polyploid contingents, indicating *Per* genes, unlike miR122 and E2F7/E2F8, are involved not only in the control of cytokinesis events, but also in events leading to nuclear polyploidy.

To explain the in vitro results, the authors argue that Desdouets and collaborators have previously demonstrated that division of binuclear hepatocytes lead to the genesis of mononuclear cells: « In hepatocyte primary culture, they demonstrated that the binuclear cells exhibited two centrosomes in G1 that were duplicated during S phase and then 4 centrosomes clustered by pairs at opposite poles of the cell during metaphase. This event led only to mononuclear 4n progeny and maintained the tetraploidy status of hepatocytes. ». Coming back into the JBC paper, Dr Desdouets demonstrated that clustering of centrosomes in binuclear hepatocytes is essential to maintain a polyploid content

and to avoid the genesis of aneuploid contingent. In fact, the binuclear contingent can either perform a normal cell cycle (genesis of polyploid mononuclear fraction) but can also fail again cytokinesis (genesis of polyploid binuclear fraction). This mechanism has been described in different papers and reviews (e.g. Duncan AW, Semin Cell Dev Biol, 2013).

REPLY:

Thank you very much for this comment. We now mention this in the corresponding section: Underlined part has been inserted in the main text.

“Thus, two cell cycles, the first failing abscission to produce binuclear cells, lead to polyploid hepatocytes with a single enlarged polyploid nucleus (Supplementary Fig. 6g, Supplementary Video 2). A similar mechanism has been previously reported to occur in insulin-induced polyploidization^{26, 55}. Occurrence of abscission failure in *Per*-null hepatocytes regardless of the previous ploidy level (inferred by nuclear size) suggests that this abscission failure is the cause of the extraordinary polyploidization observed *in vivo* (2 x 8n, 1 x 16n, 1 x 32n).”

REVIEWERS' COMMENTS:

Reviewer #4

In my opinion the concerns of reviewer 2 should NOT preclude publication of these very original studies for the following reasons. The current view in the field concerning the induction of developmentally-programmed liver cell polyploidization is that during weaning of mice hepatocytes become first binucleated (cellular ploidy $2 \times 2C$) through a cytokinesis failure. Subsequently when a binucleated cell ($2 \times 2C$) enters another cell cycle and replicates its DNA ($2 \times 4C$) and progress normally through mitosis and cytokinesis, two daughter cells are formed each containing a single enlarged nucleus ($4C$, nuclear ploidy) . So this nuclear ploidy can be a consequence of cellular ploidy when binucleated cells progress normally through the cell cycle. The group of Dr. Desdouets have shown that after weaning of wildtype mice first binucleated hepatocytes appear in the liver and afterwards mononucleated polyploid hepatocytes appear. The group of Okumura have demonstrated that this developmentally programmed polyploidization is accelerated in *Per1,2,3* resulting in the formation of more binucleated cells and mononucleated polyploid cells.

However there is also some evidence that mononucleated polyploid hepatocytes may be derived directly from mononucleated diploid cells through a process called endoreplication. Cells replicate then their DNA without progression through mitosis, so they progress through G1-S-G2 and back to G1 again. If ko of *Per1,2,3* leads to increase endoreplication the authors should have been able to observe this during their live cell imaging studies, namely the increase of nuclear size without progressing through mitosis. Endoreplication of hepatocytes has been described more under pathological conditions such as non-alcoholic fatty liver disease or liver cancer.

Since the Okumura group clearly demonstrates that *Per1,2,3* ko leads to increase binucleation during the developmental programmed liver cell polyploidization, the enhanced occurrence of cytokinesis failure in these *Per* deficient hepatocytes is in my opinion the most likely explanation for the increase in cellular and nuclear ploidy.

Point-by-point reply to Reviewers:

1. Reviewer #4:

In my opinion the concerns of reviewer 2 should NOT preclude publication of these very original studies for the following reasons. The current view in the field concerning the induction of developmentally-programmed liver cell polyploidization is that during weaning of mice hepatocytes become first binucleated (cellular ploidy 2x2C) through a cytokinesis failure. Subsequently when a binucleated cell (2 x 2C) enters another cell cycle and replicates its DNA (2x 4C) and progress normally through mitosis and cytokinesis, two daughter cells are formed each containing a single enlarged nucleus (4C, nuclear ploidy) . So this nuclear ploidy can be a consequence of cellular ploidy when binucleated cells progress normally through the cell cycle. The group of Dr. Desdouets have shown that after weaning of wildtype mice first binucleated hepatocytes appear in the liver and afterwards mononucleated polyploid hepatocytes appear. The group of Okumura have demonstrated that this developmentally programmed polyploidization is accelerated in Per1,2,3 resulting in the formation of more binucleated cells and mononucleated polyploid cells.

However there is also some evidence that mononucleated polyploid hepatocytes may be derived directly from mononucleated diploid cells through a process called endoreplication. Cells replicate then their DNA without progression through mitosis, so they progress through G1-S-G2 and back to G1 again. If ko of Per1,2,3 leads to increase endoreplication the authors should have been able to observe this during their live cell imaging studies, namely the increase of nuclear size without progressing through mitosis. Endoreplication of hepatocytes has been described more under pathological conditions such as non-alcoholic fatty liver disease or liver cancer.

Since the Okumura group clearly demonstrates that Per1,2,3 ko leads to increase binucleation during the developmental programmed liver cell polyploidization, the enhanced occurrence of cytokinesis failure in these Per deficient hepatocytes is in my opinion the most likely explanation for the increase in cellular and nuclear ploidy.

Reply: We very much appreciate this reviewer's evaluation of our work. Accepting his/her proposal, we added the following paragraph in DISCUSSION, as follows.

Hepatic polyploidization occurs mainly during liver development. In rodents, hepatocytes are exclusively diploid (2n) in neonates, polyploidization starting after weaning. Up to 90% of rat/mouse hepatocytes become polyploidy at adult age^{24,58}. The current view in the field concerning the induction of developmentally-programmed liver cell polyploidization is that during weaning of mice hepatocytes become first binucleated (cellular ploidy 2 x 2n) through a cytokinesis failure. Subsequently when a binucleated cell (2 x 2n) enters another cell cycle and replicates its DNA and progress normally through mitosis and cytokinesis, two daughter cells are formed each containing a single enlarged nucleus (4n, nuclear ploidy). Thus, nuclear

ploidy can be a consequence of cellular ploidy. This idea is in accordance with previous observations by Desdouets⁵⁹ that in wild-type mice after weaning, first binucleated hepatocytes appear in the liver, followed afterwards by mononucleated polyploid hepatocytes. Here we have demonstrated that this developmentally programmed polyploidization is accelerated in deleted with Period genes resulting in the formation of more binucleated cells and mononucleated polyploid cells.

However there is also some evidence that mononucleated polyploid hepatocytes may be derived directly from mononucleated diploid cells through a process called endoreplication^{16,60}. Cells replicate then their DNA without progression through mitosis, so they progress through G1-S-G2 and back to G1 again. However, in our case of Per-null induced polyploidy, we cannot observe endoreplication by live cell imaging of hepatocyte culture: namely we cannot observe any instance showing the increase of nuclear size without progressing through mitosis. Endoreplication of hepatocytes has been described more under pathological conditions⁶⁰ such as non-alcoholic fatty liver disease or liver cancer.